# S100A4+ macrophages facilitate zika virus invasion and persistence in the seminiferous tubules via interferon-gamma mediation

Wei Yang[1], Yan-Hua Wu[1], Shuang-Qing Liu[2], Zi-Yang Sheng[1], Zi-Da Zhen[1], Rui-Qi Gao[3], Xiao-Yun Cui[1,4], Dong-Ying Fan[1], Zhi-Hai Qin[2], Ai-Hua Zheng[5], Pei-Gang Wang🄳[1] *, Jing An🄳[1,6] *

1 Department of Microbiology, School of Basic Medical Sciences, Capital Medical University, Beijing, China, 2 Institute of Biophysics, Chinese Academy of Science, Beijing, China, 3 Department of Biochemistry and Molecular Biology, School of Basic Medical Sciences, Capital Medical University, Beijing, China, 4 Department of Science and Technology, Capital Institute of Pediatrics, Beijing, China, 5 Institute of Zoology, Chinese Academy of Science, Beijing, China, 6 Center of Epilepsy, Beijing Institute for Brain Disorders, Beijing, China

* pgwang@ccmu.edu.cn (PGW); anjing@ccmu.edu.cn, 15810200117@163.com (JA)

**Data Availability Statement:** All relevant data are within the manuscript and its Supporting Information files.

## Abstract

Testicular invasion and persistence are features of Zika virus (ZIKV), but their mechanisms are still unknown. Here, we showed that S100A4+ macrophages, a myeloid macrophage subpopulation with susceptibility to ZIKV infection, facilitated ZIKV invasion and persistence in the seminiferous tubules. In ZIKV-infected mice, S100A4+ macrophages were specifically recruited into the interstitial space of testes and differentiated into interferon-γ-expressing M1 macrophages. With interferon-γ mediation, S100A4+ macrophages down-regulated Claudin-1 expression and induced its redistribution from the cytosol to nucleus, thus increasing the permeability of the blood-testis barrier which facilitated S100A4+ macrophages invasion into the seminiferous tubules. Intraluminal S100A4+ macrophages were segregated from CD8+ T cells and consequently helped ZIKV evade cellular immunity. As a result, ZIKV continued to replicate in intraluminal S100A4+ macrophages even when the spermatogenic cells disappeared. Deficiencies in S100A4 or interferon-γ signaling both reduced ZIKV infection in the seminiferous tubules. These results demonstrated crucial roles of S100A4+ macrophages in ZIKV infection in testes.

## Author summary

Zika virus (ZIKV), a flavivirus usually transmitted by mosquito bites, was recently reported to establish long-term infection in testes and consequently to transmit sexually from male to female. To uncover the underlying mechanisms, we characterized the gene expression profile of ZIKV-infected mouse testes and selected S100A4+ macrophages as crucial factors for long-term ZIKV infection in testes. S100A4+ macrophages originate from bone marrow and are susceptible to ZIKV infection. Using ZIKV susceptible mice, we found that ZIKV infection attracted S100A4+ macrophages to accumulate in the testes

**Funding:** This work was supported by the grant from the National Natural Science Foundation of China (NSFC) (http://www.nsfc.gov.cn/). NSFC grant 81871641 to P.G.W.; NSFC grant 81972979, NSFC grant 81671971 and NSFC grant U1902210 to J.A.; NSFC grant 81902048 to Z.Y.S.; NSFC grant 81772172 to H.C.; NSFC grant U1602223 to H.N.Z. This work was also supported by the Scientific Research Plan of the Beijing Municipal Education Committee (http://jw.beijing.gov.cn/) (KM201710025002) to P.G.W. and Key Project of Beijing Natural Science Foundation B (KZ201810025035) to J.A. The funders had no role in study design, data collection and analysis, decision to publish, or preparation of the manuscript.

**Competing interests:** The authors have declared that no competing interests exist.

and differentiate into interferon-γ-expressing cells. In further experiments, we introduced S100A4+ macrophages-depleted mice and interferon-I/II signaling deficient mice. We demonstrated that interferon-γ secreted by S100A4+ macrophages induced the tight junction protein Claudin-1 to translocate from the plasma membrane into nuclei, thus increasing the permeability of the blood-testis barrier, an indispensable structure surrounding the seminiferous tubules and protecting spermatogenic cells inside from viral infection and immune attack. Taken together, we proposed a mechanism for the long-term ZIKV infection, in which S100A4+ macrophages not only function as Trojan horses to bring ZIKV into the seminiferous tubules, but also serve as a solid shelter for ZIKV replication even when spermatogenic cells have been largely destroyed by ZIKV infection.

## Introduction

Zika virus (ZIKV) belongs to the *Flaviviridae* family and *Flavivirus* genus, and is an agent for a variety of symptoms in human ranging from mild fever to microcephaly and Guillain-Barre syndrome [1–3]. Flaviviruses are usually transmitted by bites from infected vectors, but unlike other flaviviruses, ZIKV is also transmitted vertically from pregnant women to fetuses [4] or sexually from men to women [5,6]. A prospective study of 184 ZIKV-infected men showed that viral RNA is commonly present in the semen (60/184) and can persists in one patient for more than 6 months [7]. Oligospermia, haematospermia, prostatitis and painful ejaculation are reported with various frequency [8]. For example, a prospective observational study of 15 male volunteers with acute ZIKV infection revealed that total sperm count was significantly reduced from a median $119 \times 10^6$ spermatozoa at day 7 to $45.2 \times 10^6$ at day 30 [9]; a cohort study including 11 symptomatic men demonstrated that 11 of them had presence of leukocytes, 10 showed haematospermia and six showed oligospermia [10]. Because of the threat to male health, testes protection is included in efficacy evaluation for development of ZIKV-specific vaccines or medicines [11–15]. Using different animal models, sexual transmission [16–18] and testicular injuries have been proved in ZIKV-infected mice [19–22] and olive baboons [23]. Testes are immune-privileged organs and consist of two separate compartments: seminiferous tubules and interstitial spaces [24,25]. A blood-testis barrier (BTB), mainly composed of the seminiferous epithelium (Sertoli cells) and tight junction, separates spermatogenic cells in the tubule lumen from immune cells in the interstitial space, which protects spermatogenic cells from immune attack [24,25] and pathogen invasion [26]. In ZIKV-infected mice, testes show obvious inflammation accompanied by structural destruction and viral antigen distribution in the seminiferous tubules [19–23], indicating that ZIKV can cross the BTB and invade the seminiferous tubules. However, the mechanism by which ZIKV penetrates the BTB and persists in the seminiferous tubules remains unknown.

In both mice and humans, a variety of testicular cells are susceptible to ZIKV infection, including macrophages in the interstitial space, Sertoli cells lining the seminiferous tubules and spermatogenic cells inside the tubule lumen [19,21,27,28]. The diversity of target cells suggests that ZIKV may use more than one way to pass through the BTB [8,29]. ZIKV has been reported to hijack monocytes/macrophages [30], its primary target cells [31,32], as Trojan horses to pass blood-tissue barriers in ZIKV-infected fetal brains [33]and placenta [34,35], but whether this occurs in testes and eyes remains unknown. Although testicular macrophages located in the interstitial space are susceptible to ZIKV infection [22], they do not enter the seminiferous tubules under physiological conditions. Their principle duties are to maintain the immunosuppression microenvironment which is crucial for production of androgens and

development of spermatogonia [36,37]. Infection in Sertoli cells, the major component of the BTB, is another potential pathway. Susceptibility of Sertoli cells to ZIKV infection has been well studied [22,38,39], and human Sertoli cells infected by ZIKV were shown to effectively release progeny virus at the abluminal surface in an *in vitro* BTB model [40]. However, in AG6 mice lacking interferon (IFN)-α/β and -γ receptors, the ZIKV infection rate of spermatogenic cells was lower than that in A6 mice which lack only the IFN-α/β receptor, whereas Sertoli cells were infected in both models [19,21,22,41], suggesting an IFN-γ dependent pathway also contributes to ZIKV invasion of the seminiferous tubules. When ZIKV enters the tubules, spermatogenic cells are the main target cells [19,21,42]. Human spermatogenic cells can support ZIKV replication *in vitro* without obvious change in cell viability [42], indicating that they are important for ZIKV persistence in the testes. Nevertheless, ZIKV must pass through the BTB before it infects spermatogenic cells. Moreover, spermatogenic cells are largely destroyed in ZIKV infected mice [19,21], and oligospermia is reported in ZIKV-infected men [9], implying injuries to spermatogenic cells *in vivo*. Some other ZIKV-susceptible cells, therefore, may exist in the seminiferous tubules to support ZIKV replication.

Here, we show that a bone marrow-derived S100A4+ macrophage subpopulation appeared in the testes of ZIKV-infected mice and were susceptible to ZIKV infection. In ZIKV-infected mice, S100A4+ macrophages differentiated into proinflammatory M1 macrophages and expressed IFN-γ. They first accumulated in the testicular interstitial space, inducing the tight junction protein Claudin-1 (CLDN1) to translocate into the nuclei of Sertoli cells and spermatogenic cells in an IFN-γ-dependent manner, and then permeated the seminiferous tubules, where they helped ZIKV to avoid the CD8+ T immunity. S100A4+ macrophages had higher viability than spermatogenic cells, which allowed them to be the main target cells for ZIKV replication when spermatogenic cells were destroyed at the late stage of infection.

## Results

### *S100a4* gene expression increased in testes from ZIKV infected mice

To determine the mechanisms underlying testicular infection of ZIKV, we analyzed the transcriptome of ZIKV-infected testes at 5 days post infection (dpi) using RNA-sequencing. Among the top genes significantly increased upon ZIKV infection, *S100a4* attracted our attention (**S1 Table**). As a calmodulin protein regulating cell mobility, S100A4 is a functional marker for a bone marrow-derived macrophage subpopulation that will be motivated when there is tissue injury in peripheral organs [43,44]. In ZIKV-infected testes, *S100a4* expression increased by 5.7-fold (**Fig 1A**). Furthermore, the expression of *Saa3*, *Ccl2* and *Ccl5*, three chemokines critical for monocyte recruitment, increased by 273.3-, 9.0- and 8.3-fold respectively (**Fig 1A**), implying potential involvement of such cells in testicular infection of ZIKV.

Compared to AG6 mice, A6 mice are more likely to survive ZIKV infection and therefore provide a longer window to study testicular damage in ZIKV-infected mice. As revealed by RT-qPCR (**S1A Fig**), ZIKV RNA in liver and lung tissue from ZIKV-infected A6 mice peaked at 7 dpi and decreased to undetectable levels at 21 dpi. ZIKV RNA in the brain showed a similar trend, but it remained detectable even at 28 dpi, possibly due to the brain's immune-privileged environment. In contrast, ZIKV RNA in the testes displayed a different pattern; it peaked at 14 dpi but not 7 dpi, and was higher than all other tissues at most timepoints. ZIKV RNA in whole blood seems to reflect the average level of the testes and other organs. These results suggested that testes were susceptible to ZIKV infection and ZIKV tended to form a persistent infection specifically in immune-privileged organs such as the testes.

To verify the expression change of S100A4, we challenged 6- to 8-week-old male A6 mice intraperitoneally with $1 \times 10^4$ ZIKV (SMGC-1 strain), and isolated the testes at 7, 14, 21 and

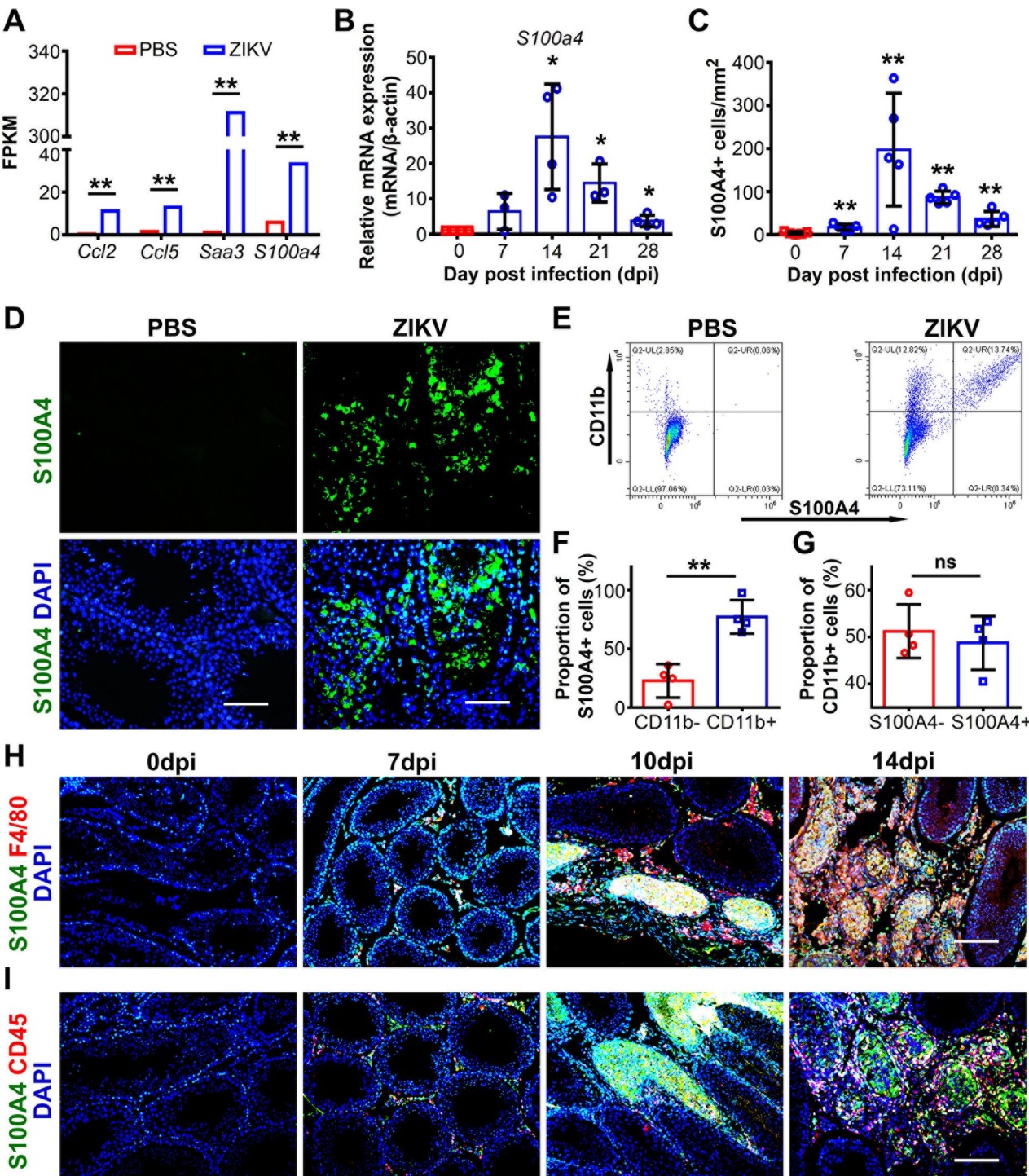

**Fig 1. S100A4+ cells in testes of ZIKV-infected mice were mainly CD11b+ CD45+ myeloid macrophage subpopulation. (A)** Transcriptome analysis of indicated genes in testes of ZIKV-infected AG6 mice at 5 dpi. Control mice were injected with PBS. (n = 3 mice for each group). **(B and C)** Dynamic changes of *S100a4* gene level and S100A4+ cells in ZIKV-infected testes. Testes from ZIKV-infected A6 mice were isolated at different time points as indicated and subjected to RT-qPCR or immunofluorescence staining (IFA) with anti-S100A4 antibody. **(B)** Change of *S100a4* gene level was expressed as relative expression to β-actin, and shown as means ± SEM. (n = 3–4 mice for each time point). **(C)** The number of S100A4+ cells were quantified by method described in the experimental procedures, and the number of S100A4+ cells were recorded as cells/mm$^2$ and was shown as means ± SEM. (n = 5 mice for each time point). **(D)** IFA for S100A4+ cells. Testes from ZIKV-infected A6 mice were collected at 14 dpi and PBS-injected A6 mice served as controls. S100A4+ cells were detected by anti-S100A4 antibody and nuclei were stained with DAPI. Scale bar, 25 μm. **(E-G)** Flow cytometry assay for S100A4+ cells. Testicular cells from ZIKV-infected (14 dpi) or PBS-injected A6 mice were subjected to flow cytometry

analysis with anti-S100A4 antibody and anti-CD11b antibody. **(E)** A representative result. **(F)** Percentage of CD11b+ cells in S100A4+ cells. Results were shown as means ± SEM. (n = 4 mice for each group). **(G)** Proportion of S100A4+ or S100A4- cells in CD11b+ cells. Results were shown as means ± SEM. (n = 4 mice for each group). **(H and I)** Co-immunofluorescence staining for S100A4+ cells. Testes from ZIKV-infected A6 mice were isolated at indicated time points and subjected to co-immunofluorescence staining with anti-S100A4 antibody and **(H)** anti-F4/80 antibody, or **(I)** anti-CD45 antibody. Nuclei were shown with DAPI. Scale bar, 50 μm. Relative mRNA expression of *S100a4* and proportion in S100A4+ cells or CD11b+ cells in ZIKV-infected testes were analyzed using the Student's t test, number of S100A4+ cells were analyzed using the Mann-Whitney U test. *p < 0.05 versus Ctrl, **p < 0.01 versus Ctrl.

28 dpi. Compared to controls injected with phosphate buffered saline (PBS), *S100a4* expression in ZIKV-infected testes, as determined by RT-qPCR, began to increase at 7 dpi, peaked at 14 dpi, and then gradually decreased and returned to almost normal level at 28 dpi (**Fig 1B**). The expression change of S100A4 in ZIKV-infected testes was further validated by immunofluorescence staining. As expected, a dramatic increase of S100A4+ cells was observed in testis sections at 14 dpi (**Fig 1C and 1D**), which was consistent with RT-qPCR results.

In addition to the testes, other organs from ZIKV-infected mice were also examined for S100A4+ cell distribution by immunofluorescence. Increased S100A4+ cells after ZIKV infection were observed only in the spleen (**S1B Fig**). Since the spleen is the macrophages reservoir [45], the result suggested that S100A4+ cells were likely to be macrophages and required development or differentiation in the spleen. Brain and lung tissues had few S100A4+ cells, and the number did not increase after ZIKV infection (**S1C and S1D Fig**). S100A4 is likely expressed by some resident cells such as microglial in the brain [46], other than myeloid macrophages recruited during infection. No other organs tested including the epididymis, liver and kidney, showed expression of S100A4 (**S1E–S1G Fig**), implying that S100A4+ cells might have unique and specific roles in ZIKV-infected testes.

## S100A4+ cells in ZIKV-infected testes were a subpopulation of myeloid macrophages

To characterize the origin of S100A4+ cells in ZIKV-infected testes, we prepared testicular cells from ZIKV-infected A6 mice, and incubated them with antibodies against S100A4 and various other cellular markers including SOX9 (Sertoli cells),α-SMA (myoid epithelial cells), DDX4 (spermatogenic cells), CD11b (macrophage), CD4 (lymphocyte) or CD8 (lymphocyte). The stained cells were then analyzed using flow cytometry. Few S100A4+ cells expressed each of the above cellular markers except for CD11b (**Figs 1E and S2A–S2E**), for which approximately 80% of S100A4+ cells were positive (**Fig 1F**), indicating that S100A4 was predominantly expressed by macrophages. Notably, although the majority of S100A4+ cells were positive for CD11b, only 48.75% of CD11b+ cells were positive for S100A4 (**Fig 1G**). The S100A4+ macrophages thus represented a subpopulation of macrophages.

S100A4+ cells in ZIKV-infected testes were further characterized by co-immunofluorescence staining. Sections from ZIKV-infected testes at 7–14 dpi were stained with anti-S100A4 antibody and anti-F4/80 or anti-CD45 antibody; the latter two are markers for macrophages or myeloid cells respectively [47,48]. S100A4 co-localized with both F4/80 or CD45, and the double positive cells increased over time before 14 dpi (**Fig 1H and 1I**). In contrast, few F4/80 positive cells were observed in testes without ZIKV infection and they did not express S100A4 (**Fig 1H**), indicating testicular resident macrophages did not express S100A4. In line with the accumulation of S100A4+ macrophages, Luminex assay confirmed the increase of CCL2, CCL3 and CCL5, which are three chemokines required for monocyte/macrophage recruitment (**S2F–S2H Fig**). Taken together, these results showed that S100A4+ macrophages in the testes were derived from bone marrow and recruited to the testes during ZIKV infection.

## S100A4+ macrophages assisted ZIKV replication in the seminiferous tubules

The results above demonstrated that, S100A4+ macrophages were distributed in both the testicular interstitial space and the seminiferous tubule lumen. Based on the finding, we hypothesized that S100A4+ macrophages could invade into the immune-privileged seminiferous tubules. To test the hypothesis, we analyzed the distribution dynamics of S100A4+ macrophages in testes at 7 to 28 dpi. Immunohistochemistry results revealed that S100A4+ macrophages were dispersed in interstitial tissue at 7 dpi, and then dramatically increased in cell number and invaded the lumen of the seminiferous tubules at 14 dpi, where they persisted for two weeks before gradually disappearing at 28 dpi (**Fig 2A and 2B**). These results indicated that S100A4+ macrophages were able to penetrate the BTB and invade into the seminiferous tubules.

In previous experiments using AG6 mice as the ZIKV infection model, we have found that F4/80+ macrophages in the testicular interstitial space were susceptible to ZIKV infection [22]. Are S100A4+ macrophages, especially those located inside the seminiferous tubules also susceptible to ZIKV? To address this question, we isolated S100A4+ macrophages from the peritoneal cavity and investigated their susceptibility to ZIKV infection. Most peritoneal macrophages were positive for S100A4 (**Fig 2C**). When they were challenged with ZIKV, ZIKV antigen was observed in most of them (**Fig 2D**), and viral RNA was detected and increased with time (**Fig 2E**), indicating a replicable infection occurred in S100A4+ macrophages. Susceptibility of S100A4+ macrophages to ZIKV infection was further investigated *in vivo*. Testis sections at 0 to 28 dpi were subjected to co-immunofluorescence staining using anti-S100A4 and anti-ZIKV antibodies. S100A4+ macrophages in both the tubule lumen and interstitial space were positive for the ZIKV antigen (**Fig 2F**). Since most S100A4+ macrophage would enter the tubule lumen, they were thought to play important roles in ZIKV replication in the seminiferous tubules.

## Intraluminal S100A4+ macrophages helped ZIKV to avoid CD8+ T immunity

Testes are well known immune-privileged organs, and immune cells are excluded from the seminiferous tubules in testes. Since S100A4+ macrophages supported ZIKV replication in the tubule lumen, do they help ZIKV evade cellular immunity? To address this question, we studied the distribution dynamics of CD8+ T cells using testis sections from ZIKV-infected mice at 7 to 28 dpi. Immunohistochemistry results showed that CD8+ T cells were mostly distributed in the interstitial space and did not enter the seminiferous tubules until 21 dpi when the integrity of the seminiferous tubules was largely destroyed (**Fig 3A**). Co-immunofluorescence staining was next used to observe whether S100A4+ macrophages contacted CD8+ T cells. Consistent with immunohistochemistry results, CD8+ T cells were scattered in the interstitial space in ZIKV-infected testes and did not co-localize with intraluminal S100A4+ macrophages (**Fig 3B**). Partial co-localization between CD8a and S100A4 was observed in the interstitial space (**Fig 3B**), further implying that the BTB protected intraluminal cells from attack by CD8 + T lymphocytes. In general, the appearance of CD8+ T in seminiferous tubules lagged behind that of S100A4+ macrophages by at least 1 week (**Fig 3C**). In line with this, most S100A4+ macrophages were not positive for caspase-3 or caspase-8, two crucial factors in exogeneous apoptosis, until 21 dpi (**S3A and S3B Fig** and **Fig 3D**). To further validate the evasion of CD8+ T lymphocytes, co-immunofluorescence staining was used to determine whether the S100A4 + macrophages or ZIKV-infected cells were positive for granzyme B (GZMB), which is the effector molecule of CD8+ T cells. As expected, while interstitial S100A4+ macrophages were

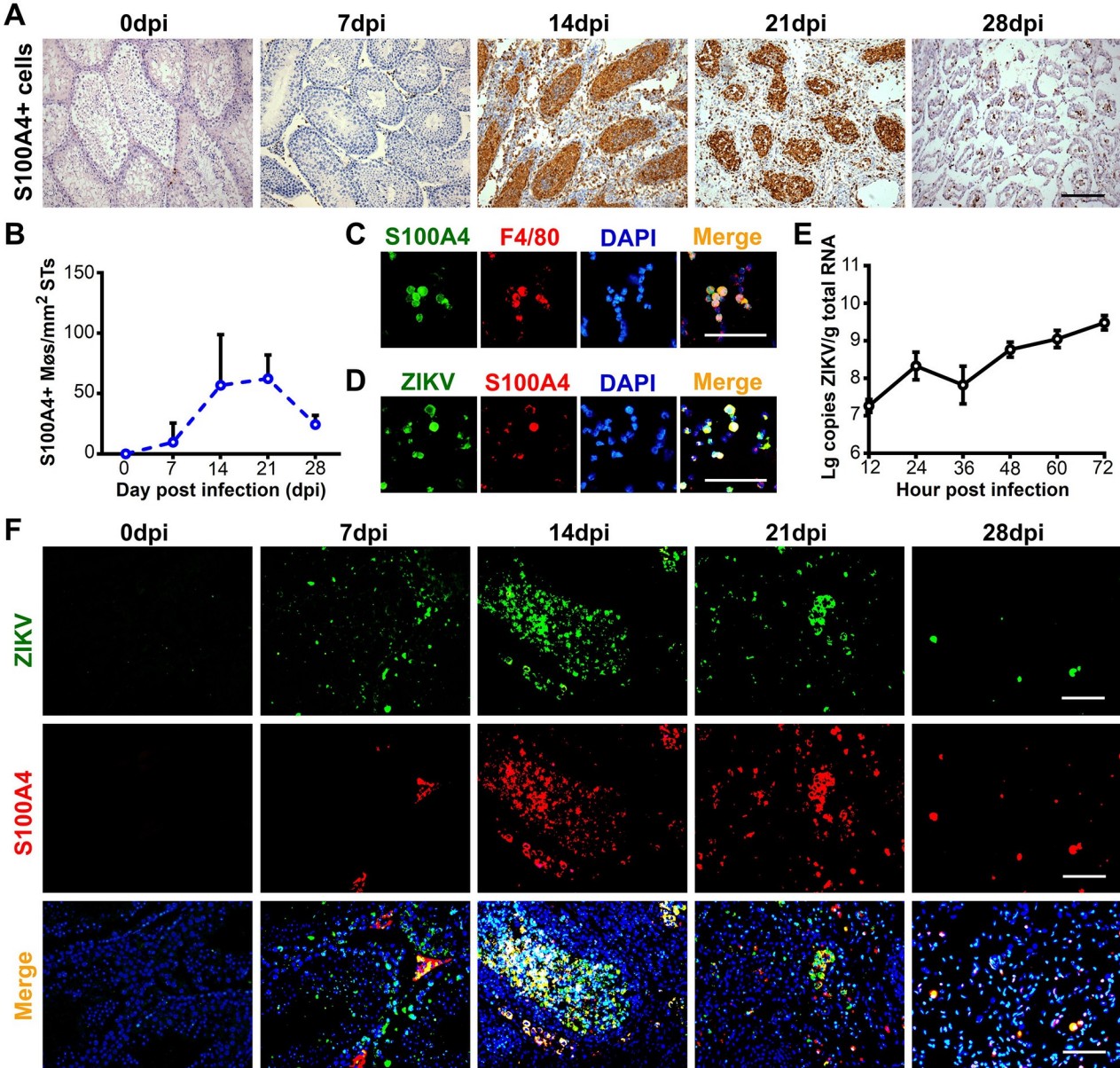

**Fig 2. S100A4+ macrophages could invade into seminiferous tubules and support ZIKV infection in the lumen of testes. (A and B)** Distribution of S100A4+ macrophages in ZIKV-infected testes. Testes from ZIKV-infected A6 mice were isolated at indicated time points and subjected to immunohistochemistry (IHC) staining with anti-S100A4 antibody. **(A)** Representative images at each time point. Scale bar, 50 μm. **(B)** Dynamic change of S100A4+ macrophages in seminiferous tubules (STs). The number of S100A4+ macrophages in seminiferous tubules were quantified by method described in the experimental procedures and expressed as cells/mm² STs, its number was shown as means ± SEM. (n = 3 mice for each time point). **(C-E)** Susceptibility of S100A4+ macrophages to ZIKV infection *in vitro*. Peritoneal macrophages were isolated as described in experimental procedures. **(C)** Peritoneal macrophages isolated from ZIKV-infected A6 mice at 7 dpi were doubly stained with anti-S100A4 and anti-F4/80 antibodies. **(D)** Peritoneal macrophages isolated from ZIKV-infected A6 mice at 7 dpi were doubly stained with anti-ZIKV and anti-S100A4 antibodies. Scale bar, 40 μm. **(E)** Peritoneal macrophages isolated from A6 mice treated with pristane were infected with ZIKV (MOI = 10), and then harvested at different time points as indicated and viral loads was determined by RT-qPCR (n = 3). **(F)** Susceptibility of S100A4+ macrophages to ZIKV infection *in vivo*. Testes from ZIKV-infected A6 mice were isolated at indicated time points and subjected to co-immunofluorescence staining with anti-S100A4 antibody and anti-ZIKV antibody. Nuclei were shown with DAPI. Scale bar, 25 μm.

positive for GZMB, most intraluminal cells were negative (**Fig 3E and 3F**). A similar result was found for ZIKV-infected cells (**Fig 3G and 3H**). As suggested by these results, while ZIKV-infected S100A4+ macrophages in the interstitial space were attacked by CD8+ T cells, those

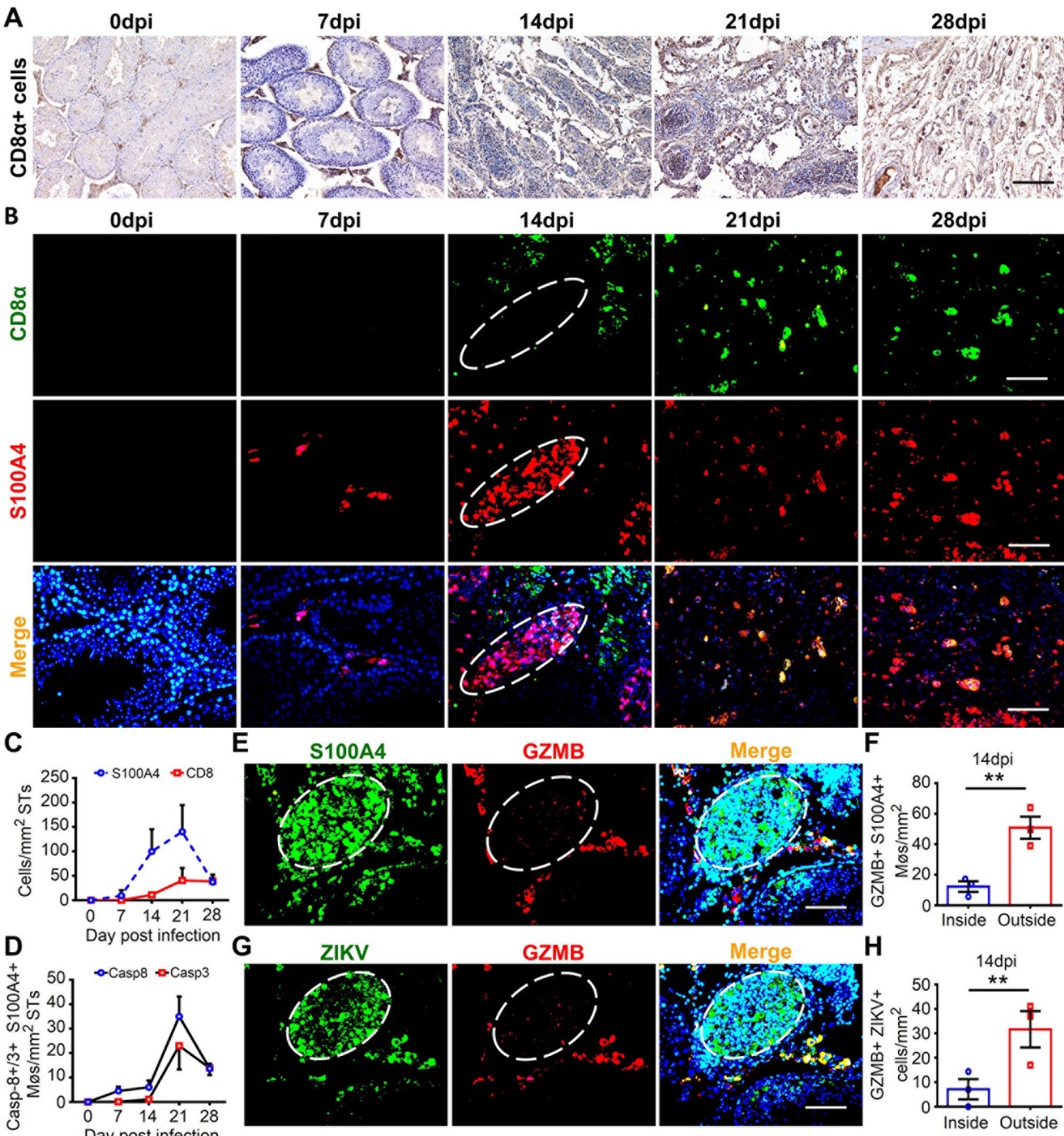

**Fig 3. Intraluminal S100A4+ macrophages were segregated from CD8+ T cells and helped ZIKV to avoid cellular immunity. (A-C)** Distribution and dynamic of CD8+ cells and their relationship with S100A4+ macrophages in ZIKV-infected testes. Testes from ZIKV-infected A6 mice were isolated at indicated time points and subjected to IHC staining with anti-CD8α antibody, Scale bar, 50 μm (**A**), or co-immunofluorescence staining with anti-CD8α and anti-S100A4 antibodies (**B**). A typical seminiferous tubule was outlined with dotted line. Scale bar, 25 μm. (**C**) The number of CD8 + cells and S100A4+ macrophages in seminiferous tubules were quantified as described in experimental procedures, and expressed as cells/mm$^2$ STs. (n = 3 mice for each time point). (**D**) The number of intraluminal S100A4+ macrophages expressing caspase-8 or caspase-3 was quantified as described in experimental procedures, and expressed as cells/mm$^2$ STs. (n = 3 mice for each time point. See S3 Fig for the representative images). **(E-H)** Expression of GZMB in S100A4+ macrophages (**E**) or ZIKV-infected cells (**G**). Testicular sections from ZIKV-infected A6 mice at 14 dpi were analyzed with co-immunofluorescence staining using anti-GZMB antibody and anti-S100A4 antibody or anti-ZIKV antibody. Scale bar, 25 μm. GZMB positive cell number of (**F**) S100A4+ macrophages or (**H**) ZIKV-infected cells were quantified inside and outside the seminiferous tubules, respectively, and expressed as cells/mm$^2$. Results were shown as means ± SEM. (n = 3 mice for each group). Number of GZMB+ S100A4+ macrophages or GZMB + ZIKV+ cells inside or outside of seminiferous tubules in ZIKV-infected testes were analyzed using the Student's t test. $^*$p < 0.05, $^{**}$p < 0.01.

in lumen stayed away from the CD8+ T cells-mediated immunity. In this way, intraluminal S100A4+ macrophages are both the target cells for ZIKV replication and the shelter for ZIKV to avoid the cellular immunity, and thereby contributing to the long-term existence of ZIKV in testes.

## S100A4+ macrophages were the predominant target cells at the late stage of ZIKV infection

Various testicular cells including macrophages, Sertoli cells and spermatogenic cells are all susceptible to ZIKV infection. We next used co-immunofluorescence staining with testicular sections obtained from ZIKV-infected mice at 7 to 28 dpi to evaluate their contribution to ZIKV infection at different time points. The ZIKV antigen was dispersed in the interstitial space and co-localized with S100A4 at 7 dpi (**Fig 2F**). Seven days later, the ZIKV antigen appeared in the seminiferous tubules, where it was mainly localized in DDX4+ spermatogenic cells and partially localized in SOX9+ Sertoli cells or S100A4+ macrophages (**Figs 4A and S4A**). Intriguingly, we observed that ZIKV utilized spermatogenic cells or S100A4+ macrophages distinctively for their replication in two adjacent seminiferous tubules (**Fig 4B**). The intraluminal distribution of ZIKV antigen did not change at 21 dpi. However, the ZIKV antigen was largely located in S100A4+ macrophages instead of DDX4+ spermatogenic cells at this time, indicating that intraluminal S100A4+ macrophages replaced spermatogenic cells to support ZIKV infection (**Fig 4C**). The reduction of spermatogenic cells was negatively correlated with an increase in S100A4+ macrophages. This suggests that their clearance may involve S100A4 + macrophages, but the detailed mechanism remains to be elucidated. In our results, co-localization of the ZIKV antigen and α-SMA was rarely observed (**S4B Fig**). The infection rate of each type of these cells was then determined by measuring the co-localization signals, and it was clear that S100A4+ macrophages replaced DDX4+ cells as the main target cells at late stage of ZIKV infection (**Fig 4C**).

To gain further insight into the process by which S100A4+ macrophages invade and persist in seminiferous tubules, we used electron microscopy to observe testis specimens at 14 dpi. Under the electron microscope, macrophages were observed in different locations including attached closely to seminiferous tubules (**Fig 4D and 4E**), accumulated in the center of the interstitial space (**Fig 4F**), scattered in the tubular lumen (**Fig 4G**) and squeezed into the seminiferous tubules (**Fig 4H**). The seminiferous tubules with macrophages attached had a nearly normal structure with differentiating spermatids observed at the other end (**Fig 4D**); however, unlike seminiferous tubules from uninfected mice (**Fig 4I**), they rarely had peritubular myeloid epithelial cells. When macrophages were squeezing the outer layer of the seminiferous tubules, spermatogenic cells had been partially damaged (**Fig 4G**). In the seminiferous tubules with macrophages scattered in the lumen, macrophages were trapped in cell debris, which looked such as dead spermatogenic cells (**Fig 4H**). Notably, although surrounded by cell debris, the macrophages in seminiferous tubules displayed a normal morphology, indicating they had strong viability. Electron microscopic observation thus provided further evidence that S100A4+ macrophages were more robust and could support ZIKV replication at the late stage of infection when spermatogenic cells diminished.

The disappearance of spermatogenic cells in our results is consistent with previous reports [21], but the mechanisms underlying this effect remain unknown. We performed co-immunofluorescence staining with anti-DDX4 and anti-caspase-3 antibodies, and found their co-localization at 21 dpi (**S5A–S5C Fig**). However, the emergence of caspase-3 staining obviously lagged behind the reduction of DDX4 staining. Moreover, by fluorescence and electron microscopy, we hardly observed the co-existence of S100A4+ macrophages and spermatogenic

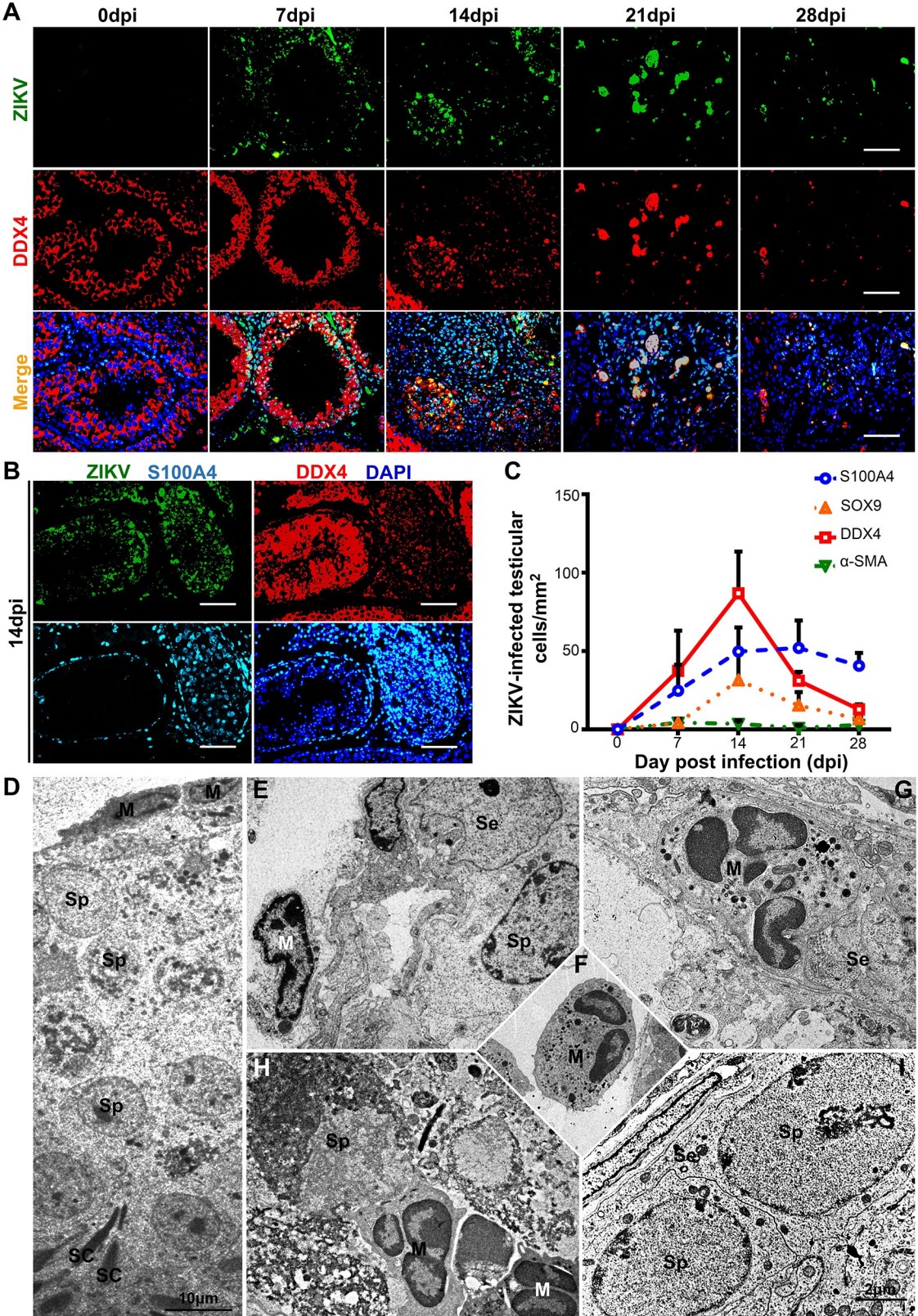

**Fig 4. Intraluminal S100A4+ macrophages replaced spermatogenic cells to support ZIKV replication at late stage of infection.** (**A**) Susceptibility of spermatogenic cells to ZIKV infection. Testes from ZIKV-infected A6 mice were isolated at indicated time

points and subjected to co-immunofluorescence staining with anti-DDX4 antibody and anti-ZIKV antibody. Nuclei were shown with DAPI. Scale bar, 25 μm. **(B)** Serial sections from ZIKV-infected testes at 14 dpi were subjected to IFA with anti-ZIKV (green), anti-DDX4 (red) or anti-S100A4 (indigo blue) antibodies. Nuclei were visualized with DAPI. Scale bar, 25 μm. **(C)** The numbers of ZIKV-infected testicular cells by type at 0–28 dpi were quantified as mentioned in experimental procedures and expressed as cells /mm². (n = 3mice for each time point. See **S4 Fig** for the representative images.) **(D-I)** Ultrastructure morphological changes of seminiferous tubules from ZIKV-infected A6 mice. **(D)** A seminiferous tubule was attached by peripheral macrophage-like cells. **(E)** Peripheral macrophage-like cells in high magnification. **(F)** Macrophage-like cells accumulated in interstitial space. **(G)** Macrophage-like cells in outer layer lining seminiferous tubules. **(H)** Intraluminal macrophage-like cells were surrounded by cell debris. **(I)** Seminiferous tubules from uninfected A6 mice. M: macrophage-like cells, Sp: spermatogenic cells, SC: sperm cells, Se: Sertoli cells.

cells. These results suggested that the death of spermatogenic cells occurred very quickly in a caspase-3 independent manner and the mechanisms need further investigation.

Taken together, these results outlined a relay among various testicular cells during ZIKV infection. S100A4+ macrophages in the interstitial space were the primary target cells. When ZIKV spread into the tubule lumen, Sertoli cells or spermatogenic cells took the second turn, with spermatogenic cells being the predominant target cells. At the late stage when spermatogenic cells are diminished, intraluminal S100A4+ macrophages replaced them to support ZIKV infection. In this way, ZIKV changed target cells and persisted in testes for longer time.

## Depletion of S100A4+ macrophages reduced ZIKV infection in spermatogenic cells

To further investigate the roles of S100A4+ macrophages during ZIKV infection in testes, we generated *S100a4⁻/⁻ Ifnar⁻/⁻* mice (SA6) by mating *S100a4* deficient mice with A6 mice (**S6A Fig**), and characterized with PCR (**S6B Fig**). When challenged with ZIKV, the body weight of SA6 mice gradually decreased after ZIKV infection, reached the lowest at 10 dpi and recovered at about 14 dpi (**S6C Fig**). No obvious difference in clinical manifestations and survival rate was found between SA6 and A6 mice (**S6D and S6E Fig**), suggesting that *S100a4* deficiency did not affect the susceptibility to ZIKV in general and that SA6 mice could serve as ZIKV infection model like A6 mice.

ZIKV RNA in whole blood, brain, liver and testis of SA6 mice was measured using RT-qPCR at 7 to 10 dpi and compared to that of A6 mice. The liver and brain of SA6 mice had a viral load change similar to A6 mice, with the peak occurring at 7 dpi (**S6F Fig**). Viral loads in the whole blood of SA6 mice were generally lower than that in A6 mice (**Fig 5A**), indicating that S100A4+ macrophages had partially contributed to ZIKV replication. Intriguingly, although no significant difference in viral load in the testis was observed between SA6 and A6 mice at 10 dpi (**Fig 5B**), ZIKV RNA in semen was significantly reduced (**S6G Fig**). Moreover, testicular viral load in SA6 mice at 7 dpi was higher than that at 14 dpi (**S6F Fig**), whereas the testicular viral load in A6 mice at 7 dpi was lower than that at 14 dpi (**S1A Fig**), implying different dynamics of ZIKV RNA in the testes in the absence of S100A4+ macrophages. These results thus suggested that bone marrow-derived S100A4+ macrophages were not required for ZIKV replication in the testes at the early stage of infection, but they played an important role in testis infection after acute infection.

The distribution of macrophages and cell tropism of ZIKV in the testes of SA6 mice was then investigated using co-immunofluorescence. As expected, S100A4 staining showed no signal in SA6 mice (**S7A Fig**). Instead, a few F4/80+ cells were scattered in the interstitial space, did not increase over time and could not enter into seminiferous tubules (**Figs S7B and 5C**), indicating that testicular resident macrophages were not able to invade the seminiferous tubules like S100A4+ macrophages. In line with this result, the ZIKV antigen in the SA6 testes was predominantly expressed by F4/80+ macrophages located in the interstitial space (**Fig 5D–5F**) as

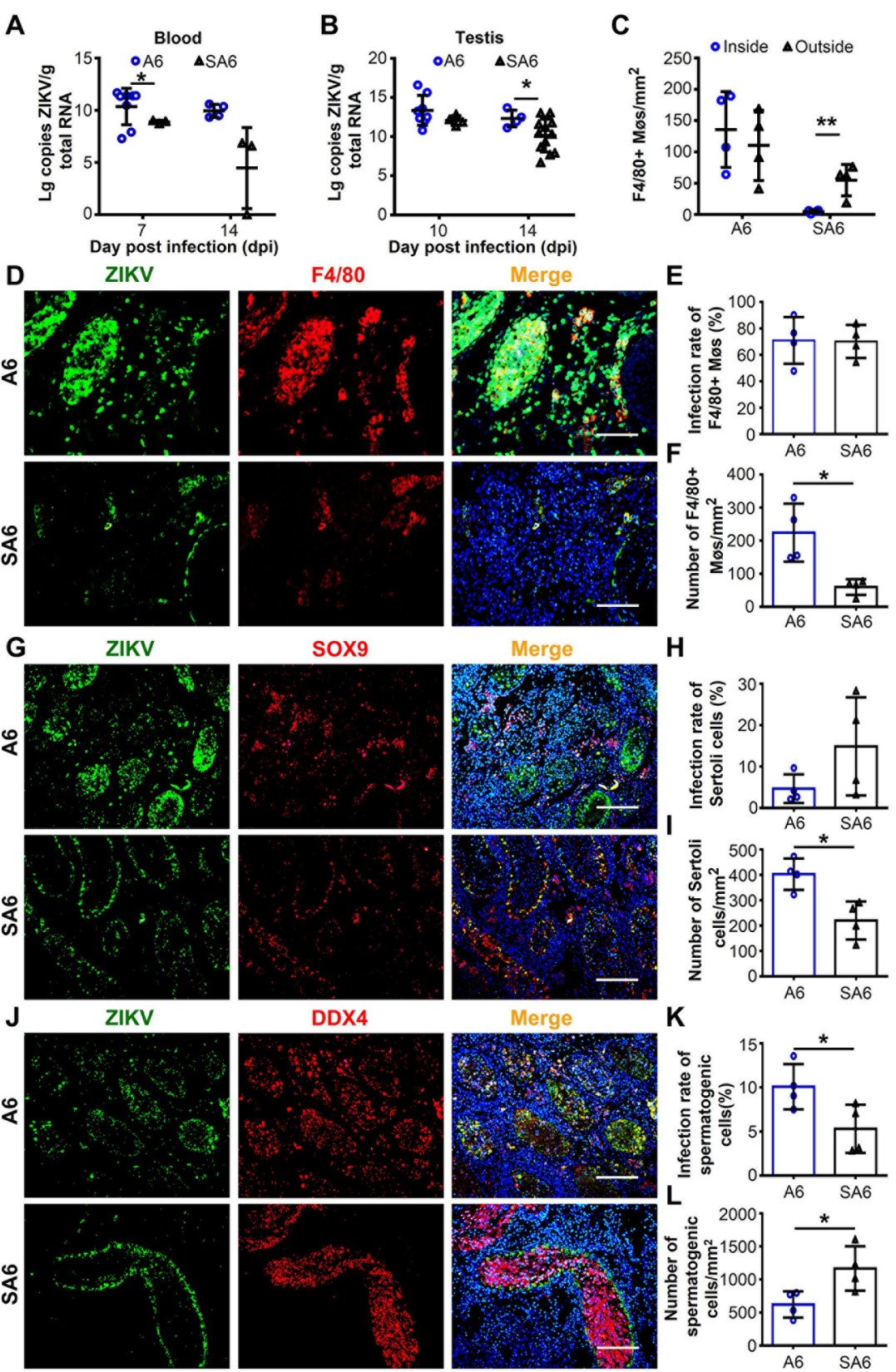

**Fig 5. Depletion of S100A4+ macrophages reduced ZIKV infection in seminiferous tubules.** SA6 mice were generated and characterized as described in **S6A and S6B Fig**. **(A and B)** Viral RNA load in **(A)** whole blood and **(B)** testes. A6 and SA6 mice were challenged with ZIKV and viral RNA load were determined by RT-qPCR. Results were shown as means ± SEM. (n = 3–12 mice for each group). **(C)** Staining intensity of F4/80+ macrophages inside and outside seminiferous tubules. A6 and SA6 mice were challenged with ZIKV and F4/80+ macrophage cells in testicular sections at 14 dpi were stained using anti-F4/80 antibody by IFA and were counted. Representative images were shown in **S7B Fig**. The staining intensity of F4/80+ macrophages inside and outside seminiferous tubules were quantified respectively and shown as means ± SEM. (n = 3 mice for each group). **(D-L)** Comparison in testicular cells targeted by ZIKV between A6 and SA6 mice. A6 and SA6 mice were challenged with ZIKV and testes were collected at 14 dpi. Testicular cells targeted by ZIKV were analyzed with co-immunofluorescence staining using anti-ZIKV antibody and antibodies against cellular marker molecules including anti-F4/80 for macrophages **(D-F)**, anti-SOX9 for Sertoli cells **(G-I)** and anti-DDX4 for spermatogenic cells **(J-L)**. **(D, G and J)** Representative images. Scale bar, 25 μm **(D)**, 50 μm **(G and J)**. **(F, I and L)** The cell intensity of indicated cell types were quantified as described in experimental procedure and were expressed as cells /mm$^2$ shown as means ± SEM. **(E, H and K)** The infection rate of indicated cell types were quantified as described in experimental procedures and were expressed as percentage of infected cells (%) shown as means ± SEM. (n = 3–4 mice for each group). Number and infection rate of cells in ZIKV-infected testes were analyzed using the Student's t test. *p < 0.05, **p < 0.01.

well in SOX9+ Sertoli cells lining the outer layer of seminiferous tubules (**Fig 5G–5I**). In contrast, fewer DDX+ spermatogenic cells were positive for ZIKV antigen (**Fig 5J–5L**), suggesting spermatogenic cells had less chance to be infected by ZIKV in the absence of S100A4+ macrophages. According to these results, S100A4+ macrophages supported ZIKV replication by themselves and also played crucial role in ZIKV invasion into the seminiferous tubules.

By using SA6 mice, we also performed immunofluorescence staining to analyze the distribution of CD8+ T cells in the absence of S100A4+ macrophages and quantified the number of CD8+ T cells inside or outside the seminiferous tubules. A number of CD8+ T cells appeared in the ZIKV-infected SA6 testes at 14 dpi; however, the majority of them accumulated in the interstitial space, and only a few could infiltrate into the seminiferous tubule (**S7C and S7D Fig**). These results demonstrated that CD8+ T cells were still unable to enter the seminiferous tubule in the absence of S100A4+ macrophages. According to these results, it was the barrier effect of seminiferous tubules, rather than the protection of S100A4+ macrophages, that blocked CD8+ T cells from entering the seminiferous tubules.

## ZIKV infection triggered CLDN1 to translocate into the nuclei of Sertoli and spermatogenic cells

Next, we focused on the mechanisms by which S100A4+ macrophages facilitated ZIKV spread into the seminiferous tubules. As the BTB is indispensable for spermatogenic cell protection, its permeability during ZIKV infection was measured using the Evans Blue (EB) leakage test. As shown by the leakage of EB, the permeability of the BTB barrier in ZIKV infected A6 mice gradually increased over time (**S8A and S8B Fig**). In further experiments, EB leakage results showed that ZIKV-infected testes had lower permeability in SA6 than in A6 mice (**S8C and S8D Fig**), indicating the involvement of S100A4+ macrophages in destroying the BTB.

Tight junction proteins including ZO1, OCLN and CLDN1, which are important components of the BTB, were analyzed using Western blot and immunofluorescence staining. Compared with controls, all the proteins showed changes in pattern or expression levels (**Figs 6A, S8E, S8F, S9A and S9B**), and the changes were more obvious in A6 mice (**S9C–S9F Fig**). Unexpectedly, CLDN1 demonstrated a dramatic change in intracellular distribution. As a transmembrane protein, CLDN1 in uninfected testes formed short filaments perpendicular to the outer edge of the seminiferous tubules (**Fig 6A**), reflecting its distribution along the plasma membrane of neighboring cells. However, the filaments disappeared in ZIKV-infected testes, and CLDN1 was redistributed from plasma membrane into the cell nucleus (**Fig 6A**). Cells showing CLDN1 in nucleus, as revealed by co-immunofluorescence staining, included both

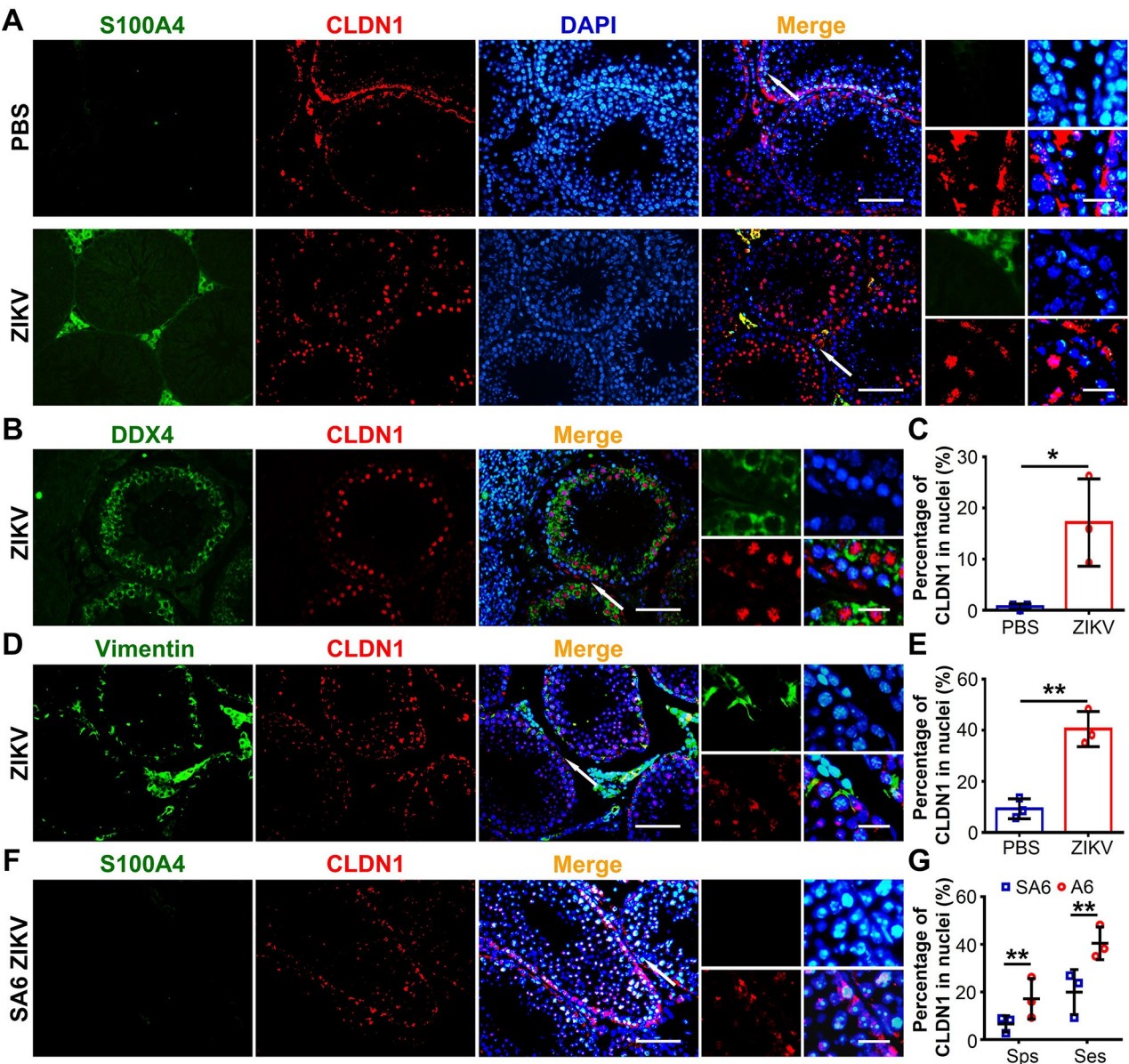

**Fig 6. ZIKV infection triggered CLDN1 to translocate into nuclei of Sertoli and spermatogenic cells.** (A-E) Testes from PBS-injected or ZIKV-infected A6 mice were collected at 7 dpi and distribution of CLDN1 were analyzed with co-immunofluorescence staining. (A) Distribution of CLDN1 in ZIKV-infected testes revealed by IFA using anti-CLDN1 and anti-S100A4 antibodies. Nuclei were stained with DAPI. Scale bar, 25 μm. (B-E) CLDN1 translocated into nuclei in various testicular cells using antibodies as indicated. (B) and (C) showing CLDN1 translocated into nuclei of spermatogenic cells (B) and their percentage in total of spermatogenic cells (Sps) (C). Scale bar, 25 μm. (D) and (E) showing CLDN1 translocated into nuclei of Sertoli cells (D) and their percentage in total of Sertoli cells (Ses) (E). Scale bar, 25 μm. All data were shown as means ± SEM. (n = 3 mice for each group). (F) Distribution of CLDN1 in ZIKV-infected testes from SA6 mice at 7 dpi. Nuclei were stained with DAPI. Scale bar, 25 μm. (G) The percentage of indicated cells with CLDN1 translocated into nuclei was quantified as described in Experimental Procedure and shown as means ± SEM. (n = 3 mice for each group). All small figures listed at the right side of the corresponding image showing area indicated by arrow in high magnification. Scale bar, 5 μm. Percentage of CLDN1 translocated into nuclei of spermatogenic cells and Sertoli cells in ZIKV-infected testes and control mice were analyzed using the Student's t test. $^{*}p < 0.05$, $^{**}p < 0.01$.

DDX4+ spermatogenic cells and vimentin+ Sertoli cells (**Fig 6B–6E**). These results suggested that ZIKV infection induced CLDN1 of spermatogenic cells and Sertoli cells to translocated from plasma membrane into nuclei and consequently increase the permeability of the BTB.

To study whether S100A4+ macrophages were involved in the translocation of CLDN1, we investigated the distribution of CLDN1 in ZIKV-infected SA6 mice. In the absence of S100A4 + macrophages, fewer cells showed CLDN1 in nucleus (**Fig 6F and 6G**), indicating that CLDN1 translocation into the nucleus required the presence of S100A4+ macrophages. Considering the reduction of ZIKV-infected spermatogenic cells and the absence of S100A4+ macrophages in the seminiferous tubules of SA6 mice, the translocation of CLDN1 into nucleus was likely a key step for ZIKV and S100A4+ macrophage invasion into the tubule lumen.

## S100A4+ macrophage-secreted IFN-γ induced CLDN1 translocation into the nucleus

As mentioned above, ZIKV-infected spermatogenic cells were less abundant in AG6 mice than in A6 mice [22]. The similarity between AG6 mice and SA6 mice led us to investigate whether IFN-γ mediated the CLDN1 translocation induced by S100A4+ macrophages. The concentration of IFN-γ in ZIKV-infected testes was measured, and it was significantly higher than in uninfected controls (**Fig 7A**). As classical activated macrophages (M1) are an important source of IFN-γ, the polarization of S100A4+ macrophages was next studied. Testis sections were stained with anti-S100A4 and anti-iNOS (M1 macrophage) or anti-CD163 (M2 macrophage) antibodies. Most S100A4+ macrophages expressed iNOS (**Figs S10A and 7B**) from 7 to 28 dpi, while only a few expressed CD163 (**S10B Fig**), showing that they polarized into the pro-inflammatory M1 type of macrophage. In further experiments using co-immunofluorescence staining, most S100A4+ macrophages were found to be positive for IFN-γ staining (**Fig 7C**), indicating that S100A4+ macrophages were the main source of this cytokine.

To further validate the M1 polarization of S100A4+ macrophages, we used the Luminex assay to measure a panel of cytokines in ZIKV-infected testes. Most pro-inflammatory cytokines including TNF-α, IL-1α, IL-12 and IL-6 were significantly increased, while anti-inflammatory cytokines such as IL-4 and IL-10 did not change or only slightly changed (**S11A–S11I Fig**). Notably, TNF-α and IL-6 peaked at 7 dpi, while IL-1α and IL-12 peaked at 14 dpi (**S11A, S11B, S11D, S11E and S11H Fig**). This suggests that testicular cells and S100A4+ macrophages, as well as other recruited immune cells, contributed to testicular inflammation during different periods, which together extended our understanding of the pathogenic mechanism in testes.

The effect of IFN-γ on the distribution of CLDN1 was then examined *in vitro* and *in vivo*. For *in vitro* experiments, Sertoli cells were treated with IFN-γ before subject to immunofluorescence staining. At 24 and 48 hours after IFN-γ treatment, CLDN1+ nuclei increased in a time dependent manner (**Fig 7D**). For *in vivo* experiments, A6 mice were injected with IFN-γ, and testis sections were subjected to immunofluorescence staining. The distribution of CLDN1 in nuclei was observed in the outer edge of the seminiferous tubules (**Fig 7E**). The *in vitro* and *in vivo* results clearly demonstrated that IFN-γ could induce CLDN1 redistribution in testes.

To confirm the role of IFN-γ, we challenged AG6 mice with ZIKV and analyzed the expression and nucleus translocation of CLDN1, the distribution of S100A4+ macrophages, and the cell tropism of ZIKV in testis sections. In the absence of IFN-γ signaling, ZIKV infection induced a smaller reduction in CLDN1 expression (**Fig 7F**). Meanwhile, S100A4+ macrophages accumulated in the interstitial space of AG6 mice and could not enter into the seminiferous tubules. Additionally, fewer cells exhibited CLDN1 in their nuclei (**Fig 7G**). In line these findings, S100A4+ macrophages in the interstitial space and SOX9+ Sertoli cells lining the tubules, rather than DDX4+ spermatogenic cells in the lumen, were positive for the ZIKV antigen (**Fig 7H and 7I**), suggesting a similar cell tropism as in SA6 mice. No difference was found

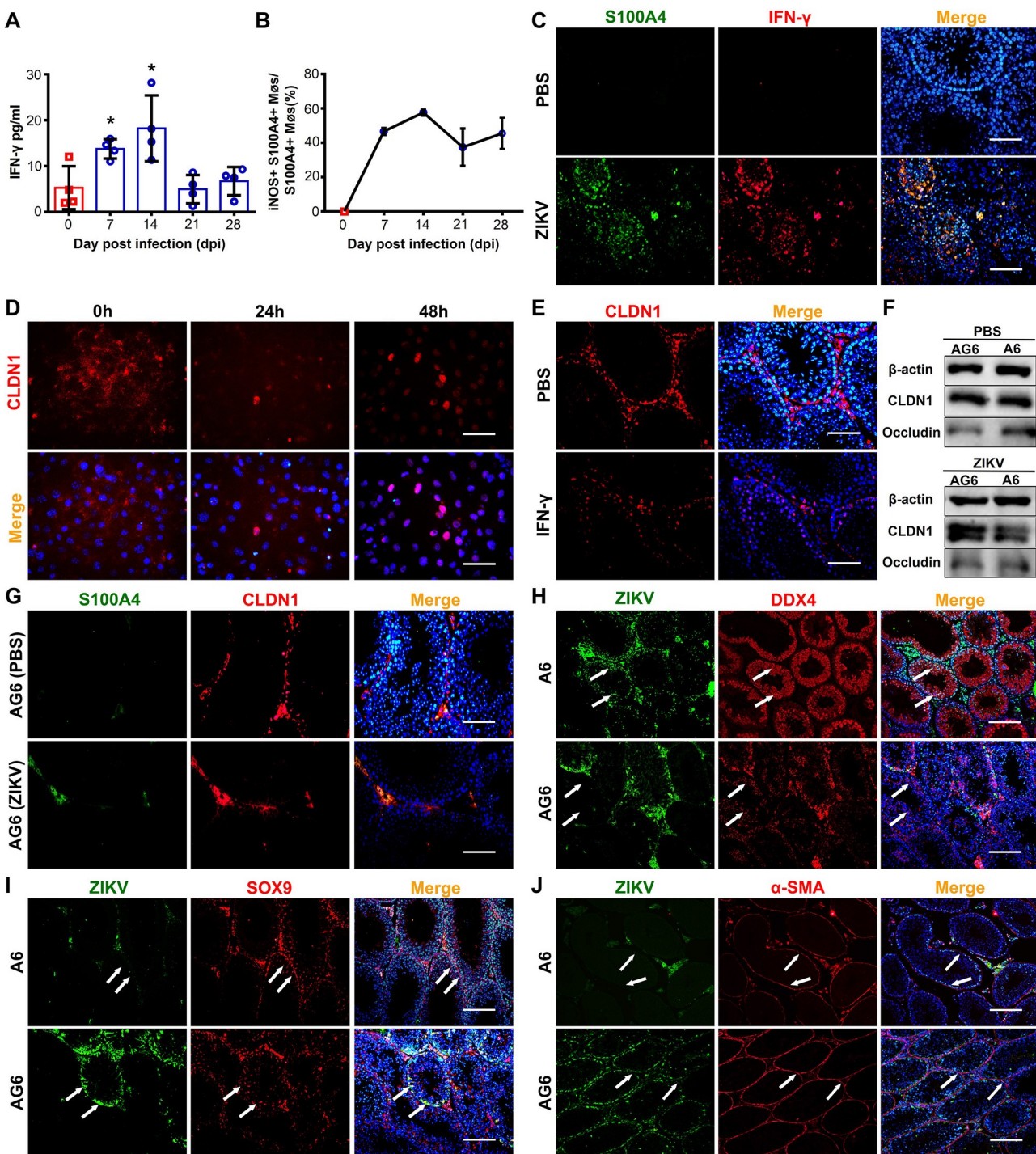

**Fig 7. S100A4+ macrophage-secreted IFN-γ induced CLDN1 translocation into cell nucleus. (A)** Dynamic of IFN-γ concentration in ZIKV-infected testes. Testes from ZIKV-infected A6 mice were isolated at indicated time points and concentration of IFN-γ in testes was measured with Bio-Plex multiplex immunoassays (n = 4 mice for each time point). **(B)** Percentage of iNOS+ S100A4+ macrophages in total S100A4+ macrophages. Testes from ZIKV-infected A6 mice were isolated at indicated time points and subjected to co-immunofluorescent staining with anti-S100A4 and anti-iNOS or anti-CD163 antibodies. See **S10 Fig** for representative images. The percentage of iNOS+ S100A4+ macrophages was quantified as described in experimental procedure. (n = 3 mice for each time point). **(C)** Expression of IFN-γ in S100A4+ macrophages. Testis sections from PBS-injected or ZIKV-infected A6 mice at 14 dpi were analyzed using co-immunofluorescent staining with anti-S100A4 and anti-IFN-γ antibodies. Nuclei were stained with DAPI. Scale bar, 25 μm. **(D)** Effect of IFN-γ on CLDN1 distribution *in vitro*. Sertoli cells were treated with 50 ng IFN-γ at 32˚C and were collected at 0, 24 and 48h. Distribution of CLDN1 was visualized by immunofluorescent staining. Nuclei were stained with DAPI. Scale bar, 25 μm. **(E)** Effect of IFN-γ on CLDN1 redistribution *in vivo*. A6 mice were intravenously injected with IFN-γ (4 μg per mouse daily) or PBS for 10 days. At 10 day after treatment, testes were

collected and distribution of CLDN1 in testicular cells was analyzed by IFA. Scale bar, 25 μm. **(F)** Expression of CLDN1 and Occludin in testes from ZIKV-infected A6 and AG6 mice as well as their corresponding controls. Mice were challenged with ZIKV or injected with PBS, and testes were isolated at 7 dpi and subjected to Western Blot. (n = 3 for each group). **(G)** Distribution of CLDN1 in ZIKV-infected AG6 testes. CLDN1 distribution in testes from PBS-injected or ZIKV-infected AG6 mice (7 dpi) were analyzed using co-immunofluorescent staining. Nuclei were stained with DAPI. Scale bar, 25 μm. **(H-J)** Susceptibility of testicular cells to ZIKV in A6 or AG6 mice. Co-localization of ZIKV antigens with various cell marker molecules in testes from ZIKV-infected A6 or AG6 mice (7 dpi) was analyzed using co-immunofluorescent staining with anti-ZIKV antibody and **(H)** anti-DDX4 antibody, **(I)** anti-SOX9 antibody or **(J)** anti-α-SMA antibody. Nuclei were stained with DAPI. Scale bar, 50 μm. IFN-γ concentration in ZIKV-infected testis tissues were analyzed using the Student's t test. *p < 0.05, **p < 0.01.

regarding ZIKV infection in α-SMA positive cells in either mouse model (**Fig 7J**). Taken together, these results showed that S100A4+ macrophages in ZIKV-infected testes secreted IFN-γ to induce the translocation of CLDN1 into nuclei of Sertoli cells and spermatogenic cells, reduced the integrity of the BTB and consequently promoted their own entry into the seminiferous tubules (**Fig 8**).

## Discussion

In this study, we used RNA-seq to analyze gene profile of ZIKV-infected testes and found that the expression of *S100a4*, a calmodulin predominantly expressed in a myeloid macrophage subpopulation [43], was dramatically increased, accompanied by similarly enhanced expression of monocyte chemokines *Saa3*, *Ccl2* and *Ccl5*. This inspired us to characterize the S100A4 + macrophages in ZIKV-infected testes, and subsequently investigate their roles in ZIKV infection. S100A4+ macrophages were recruited to the interstitial space of ZIKV-infected testes at the early stage, differentiated into IFN-γ-expressing M1 macrophages and were highly susceptible to ZIKV infection. By inducing the translocation of CLDN1 into nuclei in Sertoli and spermatogenic cells, S100A4+ macrophages increased the permeability of the BTB and subsequently promoted the invasion of ZIKV and themselves into the seminiferous tubules.

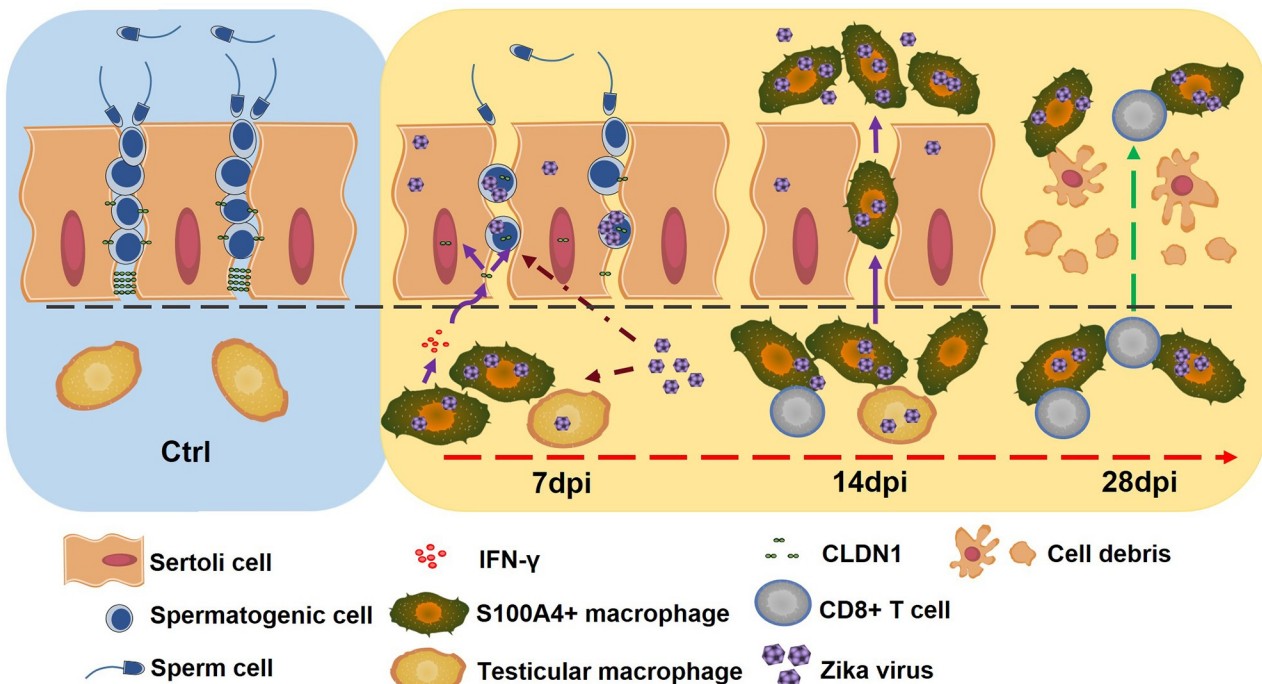

**Fig 8. Schematic diagram to show that S100A4+ macrophages assist ZIKV to infect mice testes and persist in seminiferous tubules.**

S100A4+ macrophages in the tubule lumen assisted ZIKV to stay away from the CD8+ T immunity. Moreover, because of the stronger viability, they replaced spermatogenic cells as the main target cells for ZIKV replication at the late stage when spermatogenic cells were largely destroyed. Our results suggested that S100A4+ macrophages played key roles in ZIKV infection in the testes and proposed a new model to understand ZIKV invasion and persistence in the seminiferous tubules.

S100A4 is a calmodulin, and interacts with a variety of intracellular proteins to regulate cell mobility in a calcium dependent manner [49]. Many cells including tumors of different origins [50], fibroblast subtypes in the heart [51], microglia in the central nervous system [46] and macrophage subpopulations derived from bone marrow [43] express S100A4 for multiple purposes such as tumor metastasis, tissue repair, and inflammation [44]. When periphery organs such as liver and lung are injured, S100A4+ macrophages are recruited to the injured site to participate in the inflammation or promote tissue repair [43,52,53]. In *S100a4* knockout mice, the loss of S100A4 results in impaired recruitment of bone marrow macrophages to sites of inflammation [54].

Involvement of S100A4+ macrophages in sexual transmission of viruses has not yet been reported. However, a few studies showed various roles of S100A4+ cell in the pathogenesis of sexually transmitted viruses. Anju et al. studied the role of epithelial mesenchymal trans-differentiation (EMT) in the development of human immunodeficiency virus (HIV)-1-associated nephropathy, and observed enhanced expression of FSP1/S100A4 in EMT from parietal epithelial cells to periglomerular cells or from tubular cells to peritubular cells [55]. More recently, Zhu et al. reported that the HBx gene of hepatitis B virus (HBV) promotes the occurrence and development of liver cancer by up-regulating the expression of S100A4; therefore, the authors believe that S100A4 is a potential therapeutic target for HBV-induced liver cancer [56].

According to our results, S100A4+ macrophages may facilitate ZIKV crossing of the BTB in multiple ways. First, S100A4+ macrophages could induce a translocation of CLDN1 into nuclei of Sertoli and spermatogenic cells through the mediation of IFN-γ. Sertoli cells lining the seminiferous tubules extend from the basal lamina to the tubule lumen, and form the basic structure of the BTB [57]. Spermatogenic cells intimately associated with Sertoli cells throughout spermatogenesis [58]. Although the structure of the BTB is not fully understood, CLDN1 translocation into nuclei will loosen the connection between these cells. Consequently, BTB permeability increases and spermatogenic cells are disconnected, which is beneficial for ZIKV to pass the BTB and access spermatogenic cells. Second, S100A4+ macrophages infected with ZIKV bring the virus into the seminiferous tubules acting as a Trojan horse. This may account for our results in SA6 and AG6 mice in which the testicular ZIKV load was significantly lower and the viral antigen had reduced distributed in the seminiferous tubules. However, in these two mouse models, we noted that a fraction of spermatogenic cells were still infected with ZIKV, which we thought was likely produced by infected Sertoli cells. Localized to the outer edge of the seminiferous tubules, Sertoli cells are susceptible to ZIKV infection [22,38,59]. Moreover, ZIKV infected human Sertoli cells efficiently released progeny virion in the abluminal surface in an *in vitro* BTB model [40]. Therefore, infection in Sertoli cells provided another pathway for ZIKV to pass through the BTB. From a structural perspective, Sertoli cells, placental endothelial cells and cerebrovascular endothelial cells are similar in establishing blood-tissue barriers. Intriguingly, both placental endothelial cells [35] and cerebrovascular endothelial cells [60,61] are also highly susceptible to ZIKV, so ZIKV uses similar mechanisms to cross the blood-tissue barriers in the placenta and brain [62].

In addition to helping ZIKV cross the BTB, S100A4+ macrophages also facilitate ZIKV persistence in testes for several reasons. First, macrophages either from human or mouse are highly susceptible to ZIKV and are the primary target cells during ZIKV infection [22,31,32].

Second, macrophages in the tubule lumen, as revealed by electron microscopy, showed strong viability and ability to support ZIKV replication even when spermatogenic cells were largely diminished. Third, macrophages in the tubule lumen temporarily stayed away from the attack of CD8+ T cell immunity. Here, the immune privileged environment established for spermatogenesis was used by ZIKV as a refuge. In addition, it has been reported that human spermatogenic cells support ZIKV replication *in vitro* without affecting cell survival [42]. Taken together, ZIKV may use spermatogenic cells for replication in the early stage or in intact seminiferous tubules while using S100A4+ macrophages in the late stage or in seminiferous tubules in which spermatogenic cells were destroyed. In this way, ZIKV replication can be maintained regardless of the presence or absence of spermatogenic cells.

In our results using SA6 mice, although testicular resident macrophages located in the interstitial space were infected with ZIKV, they could not invade into seminiferous tubules. Therefore, promoting ZIKV invasion of seminiferous tubules was a unique feature of S100A4 + macrophages, and testicular resident macrophages did not have such ability. Tissue resident macrophages in the testes are categorized into two distinct populations with different locations and origins: interstitial macrophages and peritubular macrophages [37]. Interstitial macrophages are embryonic-derived, associate with Leydig cells and establish an immunosuppressive microenvironment by expressing interleukin-10 to support testosterone production. Peritubular macrophages are bone marrow derived, localize in close proximity to the BTB and have tolerating functions by presenting spermatogonia auto-antigens to protect them from auto-immunity [63]. Both of these macrophages are anti-inflammatory M2 macrophages [64]. M2 macrophages have relatively weak ability to eliminate intracellular pathogens; additionally, they do not secrete IFN-γ similar to S100A4+ macrophages, so they could not induce translocation of CLDN1 and thus could not promote ZIKV invasion into the tubule lumen. Furthermore, they are significantly less abundant than S100A4+ macrophages in ZIKV-infected testes, so they not likely to be the main target cells supporting ZIKV replication.

The co-presence of M1 macrophages (S100A4+ macrophages) and M2 macrophages (testicular resident macrophages) raised a question. How do S100A4+ macrophages overcome the immune suppressive properties of resident macrophages during ZIKV infection? Immunofluorescence staining for anti-iNOS or anti-CD163 antibodies have showed that M1 macrophages were much more abundant than M2 macrophages in ZIKV-infected testes. In line with this, ZIKV-infected testes produced much more pro-inflammatory cytokines than anti-inflammatory cytokines. Based on the result, we believed that the abundance of M1 macrophages changed the M1/M2 balance in testes and consequently altered the immune suppressive testicular environment.

Although S100A4+ macrophages assisted ZIKV to cross BTB, similar to macrophages in the placenta and fetal brain, the mechanisms in the testes are largely different. Hofbauer cells, which mediate ZIKV crossing of the placental barrier [34,35], are placental villous resident macrophages. They originate from the fetus, usually differentiate into M2 macrophages, and play an important role in placental development including vasculogenesis and angiogenesis. To date, there is no evidence that Hofbauer cells develop the M1 polarity phenotype that is able to kill intracellular microbes, and they are used by various pathogens for vertical transmission to the fetus [65]. Macrophages that mediate ZIKV passing through the human fetal blood-brain barrier (BBB) are monocytes of peripheral blood. Since most of them express CD163 and do not induce strong inflammation at infection sites, they are very likely anti-inflammatory M2 macrophages [33]. In contrast, S100A4+ macrophages described in our work are M1 macrophages. In line with the pro-inflammatory polarization, ZIKV-infected testes underwent severe pathological changes accompanied by damage to the seminiferous tubules. M1 macrophages are characterized by the induction of proinflammatory mediators

and strong microbicidal activity. Hence, it is more difficult [66], at least theoretically, for pathogens to survive the intracellular environment in M1 macrophage. This may account for the fact that manifestations in the human male reproductive system are not as obvious as other symptoms.

Identification of S100A4+ macrophages provides a novel therapeutic target for ZIKV infection, especially for its sexual transmission. Various strategies targeting S100A4 or IFN-γ have been developed and tried for diseased treatment. For example, niclosamide, an established antihelminthic drug, has been proved to inhibit the transcription of S100A4 [67]. Stephen et al. tested its effects on human colorectal cancer *in vitro* and *in vivo*, and showed that HMGA2-overexpressing colorectal cancer cells were more sensitive to niclosamide [68]. Alessia et al. investigated the roles of S100A4 in microglia in animal models of amyotrophic lateral sclerosis and found that S100A4 might be a marker of microglial reactivity, and that niclosamide could control and attenuate the reactive phenotypes of microglia [46]. Based on these data, it will be interesting to test the effects of these strategies including niclosamide in ZIKV infection.

Our results were obtained from immunodeficient mice that could not completely mimic the process of ZIKV infection in humans. The persistence of ZIKV in the blood, rather than other peripheral organs, indicates that it may be more difficult for the immunodeficient mice to eliminate ZIKV from the blood. Therefore, although the immunodeficient mice were widely used for ZIKV investigation, its obvious limitation is the delayed clearance of the virus in vivo especially in these immune-privileged organs such as brain and testis. Matthew et al. had generated an immunocompetent mouse model of ZIKV infection by introducing human STAT2 into the mouse Stat2 locus (hSTAT2 KI), subcutaneous inoculation of pregnant hSTAT2 KI mice with ZIKV-Dak-MA resulted in spread to the placenta and fetal brain [69]. Investigating the mechanism of ZIKV-induced testicular damage in immunocompetent mice such as hSTAT2 KI mice will be an important part of our future work. Recently, macrophage invasion in the seminiferous tubules has also been reported in ZIKV infected male olive baboons [23]. Characterization of the origin and role of these macrophages might be helpful to further understand the contribution of S100A4+ macrophages to ZIKV infection in testes. Additionally, Asian ZIKV strains target CD14+ blood monocytes and induce M2-skewed immunosuppression during pregnancy in humans [31]; however, Asian strains used in our experiments induced M1 polarization of S100A4+ macrophages. What mechanism causes different polarization of macrophages, and how does ZIKV survive in S100A4+ macrophages which have strong microbicidal activity? The answers to these questions may be helpful for an in-depth understanding of ZIKV pathogenesis.

## Materials and methods

### Ethics statement

All animal experiments were conducted in reviewed and approved by the Institutional Animal Care and the Animal Ethics Committees of Capital Medical University (Approval number: AEEI-2015-049).

### Cells

Vero cells (African Green Monkey Kidney Cells) were cultured in minimum essential medium (MEM, Gibco, USA) with 5% fetal bovine serum (FBS, PAN, Germany) at 37˚C. C6/36 cells (Aedes albopictus cells) were cultured in RPMI 1640 (Gibco, USA) with 10% FBS at 28˚C. Sertoli Cells were purchased from China Infrastructure of Cell Line Resources and cultured in Dulbecco's Modified Eagle Medium (DMEM, Gibco, USA) with 10% FBS at 32˚C.

## Virus

Viral stocks of ZIKV (strain SMGC-1, GenBank accession number: KX266255) was propagated in C6/36 cells, this strain was isolated in Shenzhen, China from an imported ZIKV patient in 2016. The titer was determined by plaque assay on Vero cell monolayer under overlay medium containing 1.0% methylcellulose. Each vial in the stock was propagated for less than 10 times to avoid accumulation of mutations and was sequenced regularly. The passage number in this study was 6 to 8.

## Mouse

AG6 mice were previously described [22]. A6 mice were provided by the Institute of Laboratory Animals Science, Peking Union Medical College. $S100a4^{-/-}$ mice were purchased from The Jackson Laboratory. All these mice were on C57BL/6 genetic background and bred under specific pathogen-free conditions at Capital Medical University. SA6 mice were generated by mating $S100a4^{-/-}$ mice with A6 mice. Genotypes of all mice were characterized according to manufacturer protocol by standard PCR.

## Virus infection in vivo

Male mouse aged 6–8 weeks was intraperitoneally (i.p.) challenged with $10^4$ pfu ZIKV and mouse administered with PBS served as controls. Blood, sera and organs of infected mice were sampled at 7, 14, 21 and 28 dpi. Body weight, disease symptoms and survival rates of mice were recorded daily till death.

## Isolation of mouse peritoneal macrophages

Peritoneal macrophages were isolated from ZIKV-infected and uninfected A6 male mice as previously described [70]. Briefly, 8 weeks old mice were euthanized at 7 dpi and disinfected with 70% ethanol, followed by i.p. injection with 5 mL of DMEM. After massaged peritoneum for 5 min, the injected DMEM in the peritoneum cavity was collected and then centrifuged at 250×g at 4°C for 8 min. The resulting cell pellet was resuspended with DMEM with 10% FBS. Isolated peritoneal macrophages were stained with anti-F4/80 antibody (1:100, Abcam, ab111101) to monitor the cell purity, or with mouse anti-ZIKV polyclonal antibody (1:1000) to detect the distribution of ZIKV antigens. The coexistence of S100A4 with ZIKV or F4/80 was examined by incubated with rabbit anti-mouse S100A4 (1:500, Cell Signaling Technology, 13018S) or mouse anti-mouse S100A4 antibodies (1:500, proteintech, 66489–1) as primary antibodies. Donkey anti-mouse IgG (1:1000, Alexa Fluor R488, Life technologies, USA, A21202) or donkey anti-rabbit IgG (1:1000, Alexa Fluor R 594, Life technologies, USA, A21207) served as secondary antibody and images were captured with Olympus microscope (IX71, Olympus, Japan). Peritoneal macrophages from uninfected A6 mice served as controls.

## Virus infection in vitro

To investigate whether S100A4+ macrophage could support replication of ZIKV, peritoneal macrophages from A6 male mice were collected after intraperitoneal treatment with pristane for 7 days. Resuspended peritoneal macrophages were seeded to 48-well plate with a cell density of $5\times10^5$ cells/ml and cultured for 2 h at 37°C. After removed non-adherent cells by washing with PBS, peritoneal macrophages were infected with ZIKV at a multiplicity of infection (MOI) of 10 at 37°C for 1 h. The cells were harvested during 12–72 h after infection and treated with Trizol (Transgen China, ET101-01), and then ZIKV mRNA was quantitated as previously described [22].

## BTB permeability assay

Permeability of BTB was assessed by measuring Evans blue concentration in testes tissue according to previous report [71]. In brief, ZIKV-infected mice were injected intravenously with 2% Evans blue dye in saline (2 ml/kg). At 1 h after the injection, mice were re-perfused with PBS to remove Evans blue in circulation and then testes were sampled under anesthesia. The samples were weighed and cut into pieces then incubated in formamide (1 ml/100 mg) at 60˚C for 24 h. The Evans blue concentration was determined by spectrophotometry at 620 nm. Quantitative calculation of the dye in testes was based on external standards dissolved in formamide (1 ml/100 mg). Testes from uninfected male mice served as controls.

## Treatment of A6 mice with IFN-γ

To investigate the effect of IFN-γ on CLDN1 in A6 mice testes, A6 male mice were intravenously given 0.4μg IFN-γ every day for 10 days. Then testes were harvested at the 10th day after treatment. Mice administered with PBS served as controls.

## Immunofluorescence Staining (IFA)

Testes of ZIKV-infected and control mice were sampled at different time point for examination of pathological changes. Testis was fixed in Modified Davidson's Fluid solution (30 ml of 40% formaldehyde, 15 ml of ethanol, 5 ml of glacial acetic acid and 50 ml of distilled water) overnight, while other organs were fixed in 4% paraformaldehyde (PFA) solution overnight, then dehydrated and paraffin-embedded. Sections (5 μm in thickness) were incubated with the following primary antibodies overnight at 4˚C, including rabbit anti-mouse CD8α (1:400, Cell Signaling Technology, D4W2Z), rabbit anti-mouse CD4 (1:200, Abcam, ab183685), rabbit anti-mouse CD45 (1:100, Abcam, ab10558), rabbit anti-Granzyme B (1:100, Abcam, ab4059), mouse anti-mouse iNOS (1:200, Abcam, ab49999), rabbit anti-mouse CD163 (1:200, Abcam, ab182422), rabbit anti-mouse S100A4 (1:500, Cell Signaling Technology, 13018S), mouse anti-mouse S100A4 (1:500, proteintech, 66489–1), rabbit anti-mouse CLDN1 (1:200, Abcam, ab15098), rabbit anti-mouse Occludin (1:200, Abcam, ab167161), rabbit anti-mouse ZO-1 (1:200, Thermo Fisher, 33–9100), rabbit mouse anti-F4/80 (1:100, Abcam, ab111101), rabbit anti-mouse Caspase-9 (1:500, Abcam, ab202068), rabbit anti-mouse Caspase-8 (1:500, Abcam, ab227430), rabbit anti-mouse Cleaved Caspase-3 (1:500, Cell Signaling Technology, 9664S), rabbit anti-mouse DDX4 (1:100, Abcam, ab13840), mouse anti-DDX4 (1:100, Abcam, ab27591), rabbit anti-mouse α-SMA (1:100, Abcam, ab5694), rabbit anti-mouse SOX9 (1:200, Abcam, AB5535), mouse anti-mouse Vimentin (1:100, Abcam, ab8978), rabbit anti-mouse IFN-γ (1:50, Abcam, ab9657) antibody or mouse anti-ZIKV polyclonal serum(1:1000). After cleaned with PBS, sections were incubated with secondary antibodies at 37˚C for 1 h, including donkey anti-mouse IgG (1:1000, Alexa Fluor R 488, A21202, Life technologies), donkey anti-rabbit IgG (1:1000, Alexa Fluor R 594, A21207, Life technologies), goat anti-rabbit IgG (1:1000, Alexa Fluor 488, A-11008, Life technologies) or goat anti-mouse IgG 594 (1:1000, Alexa Fluor, A-11005, Life technologies). Images were captured with Olympus microscope (IX71, Olympus, Japan).

## Cell quantification in IFA image

Positive staining cells for antibodies as above indicated in IFA images were analyzed with Image J software. ZIKV infection group or control group contained at least three mice. A section from each mouse was analyzed in at least five fields (200×) and each field contained more than 1000 cells. Cell number was expressed as cells/mm$^2$. The expression of ZO-1 and

Occludin in the testes from ZIKV-infected mice at 14 dpi were also analyzed by calculating the area of red as a percentage of the total area by Image J.

## Immunohistochemistry staining (IHC)

To detect the distribution of S100A4+ cells and CD8+ T cells in testes, testes sections were incubated with rabbit anti-mouse CD8α (1:400, Cell Signaling Technology, D4W2Z) or rabbit anti-mouse S100A4 (1:500, Cell Signaling Technology, 13018S) antibody as primary antibody overnight at 4˚C. After washed with PBS, sections were stained with a secondary HRP-conjugated goat anti-rabbit antibody (Zhongshan Golden Bridge Bio Co., Ltd., China) for 1h. The reaction was visualized by addition of 3, 30-diaminobenzidine (DAB) as chromogenic reagent and stopped by removing DAB.

## Transmission Electron Microscope (TEM)

Testes of ZIKV-infected or control A6 mice were fixed with 2.5% glutaraldehyde in phosphate buffer (PB) for 2h and postfixed with 1% OsO4 in PB for 1h. After washed three times in PB, specimens were dehydrated by a graded series of ethanol. Successively infiltrated in 1:1 mixture of absolute acetone and embedding medium for 1 h and followed by 1:3 mixture of absolute acetone and embedding medium for 3 h. Then the specimens were subjected to embedding medium overnight. After embedded and ultrathin sectioned, sections were stained by uranyl acetate and alkaline lead citrate. Images were captured with transmission electron microscope (H-7500, HITACHI, Japan).

## ZIKV mRNA quantification and S100A4 mRNA relative quantification

ZIKV-infected and control mice were euthanized and whole blood and major organs were harvested at different time points as indicated. Samples were homogenated in Trizol (Transgen China, ET101-01) and RNA was extracted according to manufacturer protocol. Real-time qPCR analyses were performed as previously reported [22] with Quant One Step qRT-PCR (Tiangen, China) on 7500 Real Time PCR System (Applied Biosystems, USA). Quantification of the copies of ZIKV mRNA was determined using the standard curve method. The ZIKV genome RNA transcribed *in vitro* was quantified and used as a standard template to establish the standard curve. Relative quantification of S100A4 mRNA was determined by using β-actin as internal control and $2^{-\Delta\Delta Ct}$ method. The primer sequences were as follows: mS100A4 forward 5′-AGCTGCATTCCAGAAGGTGA-3′, mS100A4 reverse 5′-CCCACTGGCAAACT ACACCC-3′.

## Flow cytometry

Testicular cells were isolated from ZIKV-infected and uninfected A6 mice testes as previously reported [72]. Briefly, weighted testicular tissues were incubated with digestion buffer containing 1 mg of collagenase/dispase, 1 mg of hyaluronidase and 1 mg of DNAse I in 1mL of DMEM/F12 and incubated at 37˚C for 25 min with slow continuous rotation, followed by fixed with 4% PFA and permeabilized with ice-cold 90% methanol. The testicular cells were then incubated with fluorochrome-conjugated antibodies to CD11b-APC (1:100, e-bioscience, 17-0112-81) or fluorochrome-unconjugated rabbit anti-mouse CD8α antibody (1:100, Cell Signaling Technology, D4W2Z), rabbit anti-mouse CD4 antibody (1:100, Abcam, ab183685), rabbit anti-mouse S100A4 antibody (1:100, Cell Signaling Technology, 13018S), mouse anti-mouse S100A4 antibody (1:100, proteintech, 66489–1), rabbit anti-mouse DDX4 antibody (1:100, Abcam, ab13840), rabbit anti-mouse SOX9 antibody (1:100, Abcam, AB5535) or rabbit

anti-mouse α-SMA antibody (1:100, Abcam, ab5694). FITC-conjugated goat anti-mouse IgG antibody (1:1000, Termo Fisher Scientifc, F2761) and donkey anti-rabbit IgG PE secondary antibody (1:1000, Bioscience, 12-4739-81e) were used as secondary antibodies. Cells types were identified based on expression of CD11b+ for monocyte-derived macrophage, S100A4 + for S100A4+ cells, CD8α+ for CD8+ T cells, CD4+ for CD4+ T cells, DDX4+ for spermatogenic cells, SOX9+ for Sertoli cells, α-SMA+ for myoid epithelial cells. Samples were processed on a DxFLEX flow cytometer (Beckman Coulter, USA) and data was analyzed using CytExpert software (version 2.0). Compensation was performed by conjugated with fluorescent antibodies for each channel.

## Western blotting

Relative expression of tight junction-related proteins of testes from ZIKV-infected and uninfected mice were measured by Western blot. Testes were harvested and lysed by radioimmunoprecipitation assay (RIPA) lysis buffer containing protease inhibitor, followed by determination of protein concentration in each sample by BCA Protein Assay kit. Then the samples were subjected to 10% sodium dodecyl sulfate polyacrylamide gel electrophoresis and transferred onto polyvinylidene difluoride (PVDF) membranes. After being blocked in 5% milk at room temperature for 1h, the membranes were incubated with rabbit anti-mouse CLDN1 (1:200, Abcam, ab15098), rabbit anti-mouse Occludin (1:200, Abcam, ab167161) or rabbit anti-β-actin mAb (1:1000, Cell Signaling Technology, 4970S) antibodies separately at 4˚C overnight, and followed by incubation with peroxidase-linked secondary antibodies. Protein expression was visualized on an Odyssey infrared imaging system (Odyssey LI-COR Biosciences, Lincoln).

## Treatment of Sertoli Cells with IFN-γ

Sertoli cells were treated with 50 ng IFN-γ at 32˚C for 0, 24 and 48 h, respectively. Cells were collected at different time point and then fixed with 4% PFA. After washed with PBS, cells were incubated with rabbit anti-mouse CLDN1 (1:200, Abcam, ab15098) polyclonal antibody at 4˚C overnight. Donkey anti-rabbit IgG (1:1000, Alexa Fluor R 594, Life technologies, USA, A21207) served as secondary antibody. Images were captured with Olympus microscope (IX71, Olympus, Japan).

## Transcriptome analysis

Testes from ZIKV-infected AG6 mice were harvested at 5 dpi. Uninfected mice served as controls. Samples were subjected to KangChen Bio-tech. Inc. (Shanghai, China) for transcriptome analysis as previously described [22]. There were three mice in each group. Image processing and base recognition were conducted by Solexa pipeline version 1.8 (Off-Line Base Caller software, version 1.8). Gene expression level and the transcription level (FPKM value) were calculated by Cufflinks 2 software (v2.1.1) to screen differentially expressed genes between groups.

## Measurement of cytokines and chemokines levels in testes

Testes from ZIKV-infected A6 mice were harvested at 7, 14, 21 and 28 dpi. Weighted testes were homogenated in RIPA lysis buffer containing protease inhibitor. After determination of protein concentration, level of cytokines and chemokines in testes was detected by the Bio-Plex multiplex immunoassays according to manufacturer's instructions (Bio-Rad, USA).

## Statistical analysis

SPSS 17.0 Software (IBM, Armonk, NY, USA) was used for statistical analysis. The quantitative data between two groups with normal distributions was analyzed using the repeated-measures analysis of variance or the Student's t test. Data with abnormal distributions of variance were analyzed by the nonparametric Mann–Whitney U test. Changes of body weight and symptom score were analyzed using repeated-measures ANOVA.

All results were presented as the mean ± standard error of the mean (SEM) in this research from at least three different repeats. $P < 0.05$ was considered as has statistically significant between two groups.

## Supporting information

**S1 Table. Up-regulated genes in testes of ZIKV-infected AG6 mice at 5 dpi.** Transcriptome of ZIKV-infected AG6 testes was analyzed by RNA-sequencing. All up-regulated genes were showed in this table.
(PDF)

**S1 Fig. Immunofluorescence assay for S100A4+ cells and ZIKV RNA loads in main organs of ZIKV-infected mice. (A)** ZIKV RNA in the whole blood, testes, brain, liver and lung obtained from ZIKV-infected A6 mice from 7 to 28 dpi was measured using RT-qPCR and shown as means ± SEM (n = 3–4 mice for each group), and analyzed using the Student's t test. $^*p < 0.05$, $^{**}p < 0.01$. **(B-G)** Male A6 mice were challenged with ZIKV or injected with PBS. Main organs including **(B)** spleen, **(C)** brain, **(D)** lung, **(E)** epididymis, **(F)** liver and **(G)** kidney were isolated at 14 dpi and analyzed using immunofluorescence staining with anti-S100A4 antibody. Nuclei were shown with DAPI. Scale bar, 25 μm. Related to **Fig 1D**.
(TIF)

**S2 Fig. Flow cytometry assay for S100A4+ cells and macrophages-recruitment related chemokines production in ZIKV-infected testes of A6 mice.** Testicular cells from ZIKV-infected (14 dpi) or PBS-injected A6 mice were subjected to flow cytometry analysis with anti-S100A4 antibody and **(A)** anti-DDX4 antibody, **(B)** anti-SOX9 antibody, **(C)** anti-α-SMA antibody, **(D)** anti-CD8α antibody, or **(E)** anti-CD4 antibody. Percentage of S100A4+ cells in each population were shown as means ± SEM. (n = 3–4 mice for each group). Related to **Fig 1E and 1F**. **(F-H)** Expression of 3 macrophages-recruitment related chemokines in ZIKV-infected testes of A6 mice from 7 to 28 dpi was measured using Luminex assay and shown as means ± SEM. (n = 3–4 mice for each time point). Concentration of chemokines were analyzed using the Student's t test. $^*p < 0.05$ versus 0 dpi, $^{**}p < 0.01$ versus 0 dpi.
(TIF)

**S3 Fig. Dynamic of caspase-3 and caspase-8 expression in ZIKV-infected testes.** Testes from ZIKV-infected A6 mice were isolated at indicated time points and subjected to co-immunofluorescence staining with anti-S100A4 antibody and **(A)** anti-caspase-8 antibody, or **(B)** anti-caspase-3 antibody. Nuclei were shown with DAPI. Scale bar, 25 μm. The quantification of these results was shown in **Fig 3D**.
(TIF)

**S4 Fig. Detection of ZIKV antigens in Sertoli cells and myoid like epithelial cells during infection.** Testes from ZIKV-infected A6 mice were isolated at indicated time points and subjected to co-immunofluorescence staining with anti-ZIKV antibody and **(A)** anti-SOX9 antibody, or **(B)** anti-α-SMA antibody. Nuclei were shown with DAPI. Scale bar, 25 μm. The

quantification of these results was shown in **Fig 4C**.
(TIF)

**S5 Fig. Dynamic of caspase-3 and DDX4 expression in ZIKV-infected testes of A6 mice.**
Testes from ZIKV-infected A6 mice were isolated at indicated time points and subjected to co-
immunofluorescence staining with anti-DDX4 antibody and anti-caspase-3 antibody **(A)**.
Nuclei were shown with DAPI. Scale bar, 25 μm. Number of DDX4+ spermatogonium cells
**(B)** and DDX4+ caspase-3+ cells **(C)** at 7–28 dpi was analyzed by Image J and shown as
means ± SEM (n = 3 mice for each group). The number of indicated cells were analyzed using
the Student's t test. $^*$p < 0.05 versus 0 dpi, $^{**}$p < 0.01 versus 0 dpi.
(TIF)

**S6 Fig. ZIKV infection in SA6 mice. (A)** SA6 mice were generated by mating S100A4 defi-
cient mice with A6 mice. **(B)** Identification of SA6 mice with genomic PCR. **(C-E)** 6–8 weeks
old SA6 male mice were i.p. challenged with $10^4$ pfu of ZIKV. Mock mice were injected with
PBS. Body weights **(C)**, symptom scores **(D)** and survival rates **(E)** were monitored daily.
(n = 6–8 mice for each group). **(F)** ZIKV RNA in whole blood and testes, brain, and liver from
ZIKV-infected SA6 mice at 7 and 14 dpi were measured using RT-qPCR and shown as
means ± SEM (n = 3–4 for each group). **(G)** ZIKV RNA in semen from ZIKV-infected A6 and
SA6 mice at 10 dpi were determined by RT-qPCR. Results were shown as means ± SEM.
(n = 3 mice for each group). Comparison in body weight and symptom score among three
groups were analyzed using repeated-measures ANOVA. $^*$p < 0.05, $^{**}$p < 0.01. ZIKV RNA
loads were analyzed using the Student's t test. $^*$p < 0.05, $^{**}$p < 0.01.
(TIF)

**S7 Fig. Distribution of F4/80+ macrophages and CD8+ T cells in testes of ZIKV-infected
SA6 at 14 dpi. (A)** Testicular sections from ZIKV-infected SA6 mice at 14 dpi were analyzed
by immunofluorescence staining with anti-S100A4 antibody. Nuclei were shown with DAPI.
Scale bar, 25 μm. **(B)** Distribution of F4/80+ macrophages in testes from ZIKV-infected A6
and SA6 mice. Testicular sections from ZIKV-infected SA6 and A6 mice at 14 dpi were ana-
lyzed by immunofluorescence staining with anti-F4/80 antibody. Nuclei were shown with
DAPI. Scale bar, 25 μm. The quantification of these results was shown in **Fig 5C**. **(C)** Distribu-
tion of CD8+ T cells in testes from ZIKV-infected SA6 mice. Testes from ZIKV-infected SA6
mice at 14 dpi were isolated and subjected to immunofluorescence staining with anti-CD8α
antibodies. The staining intensity of CD8+ cells inside and outside seminiferous tubules were
quantified respectively and shown as means ± SEM **(D)**. (n = 3 mice for each group). Nuclei
are visualized with DAPI. Scale bar, 25 μm. Data were analyzed using the Student's t test.
$^*$p < 0.05, $^{**}$p < 0.01.
(TIF)

**S8 Fig. Changes of BTB permeability and CLDN1 in ZIKV-infected testes. (A and B)**
Changes of BTB permeability in ZIKV-infected testes of A6 mice. ZIKV-infected A6 mice
were injected with Evans blue (EB) at indicated time point. **(A)** Representative pictures of tes-
tes injected with EB. **(B)** EB concentration in testicular tissues was determined by spectropho-
tometry at 620 nm and shown as means ± SEM. (n = 3–6 mice for each time point). **(C and D)**
ZIKV-infected A6 and SA6 male mice were injected with EB at 0 and 14 dpi. **(C)** Representa-
tive pictures of testes injected with EB. **(D)** EB concentration in testicular tissues was deter-
mined by spectrophotometry at 620 nm and shown as means ± SEM. (n = 3–4 mice for each
time point). EB concentration in testicular tissues from ZIKV-infected A6 or SA6 male mice
was analyzed using the Student's t test. $^*$p < 0.05, $^{**}$p < 0.01. **(E)** Expression of CLDN1 in
ZIKV-infected testes of A6 mice. Testes from PBS-injected or ZIKV-infected (14 dpi) A6 mice

was subjected to Western blot with anti-CLDN1 or anti-β-actin antibody. **(F)** Fold change of CLDN1 was analyzed by Image J and shown as means ± SEM. (n = 3 mice for each group) and analyzed using the Student's t test. $^*p < 0.05$, $^{**}p < 0.01$.
(TIF)

**S9 Fig. Changes of tight junction related proteins in ZIKV-infected testes from A6 and SA6 male mice. (A and B)** Analysis of tight junction related proteins in ZIKV-infected testes. Testicular sections from ZIKV-infected A6 mice at indicated time points were analyzed by co-immunofluorescence staining with anti-S100A4 antibody and **(A)** anti-ZO-1 antibody or **(B)** anti-Occludin antibody. Nuclei were stained with DAPI. Scale bar, 25 μm. Related to **Fig 6**. **(C and D)** Testicular sections from ZIKV-infected A6 and SA6 mice at 14 dpi were analyzed by co-immunofluorescence staining with anti-ZIKV antibody and anti-ZO-1 antibody **(C)** or anti-Occludin antibody **(D)**. Nuclei were stained with DAPI. Scale bar, 25 μm. **(E and F)** The expression of ZO-1 **(E)** and Occludin **(F)** in testes from ZIKV-infected A6 or SA6 male mice at 14 dpi were analyzed by calculating the area of red as a percentage of the total area by Image J and shown as means ± SEM (n = 3 mice for each group). Data were analyzed using the Student's t test. $^*p < 0.05$, $^{**}p < 0.01$.
(TIF)

**S10 Fig. Characterization of S100A4+ macrophage polarization in ZIKV-infected testes.** Testes from ZIKV-infected A6 mice were isolated at indicated time points and subjected to co-immunofluorescence staining with anti-ZIKV antibody and **(A)** anti-iNOS antibody, or **(B)** anti-CD163 antibody. Nuclei were shown with DAPI. Scale bar, 25 μm. The quantification of these results was shown in **Fig 7B**.
(TIF)

**S11 Fig. Cytokines production in ZIKV-infected testes of A6 mice from 7 to 28 dpi.** Expression of various cytokines in ZIKV-infected testes of A6 mice from 7 to 28 dpi was measured using Luminex assay and shown as means ± SEM. (n = 3–4 mice for each time point). All data were analyzed using the Student's t test. $^*p < 0.05$ versus 0 dpi, $^{**}p < 0.01$ versus 0 dpi.
(TIF)

## Acknowledgments

We are grateful to Prof. Dai-Shu Han (Institute of Basic Medical Sciences, Chinese Academy of Medical Sciences), Prof. Cheng-Feng Qin (Beijing Institute of Microbiology and Epidemiology) and Prof. Gong Cheng (Tsinghua University, China) for their constructive advice on research design and refinement.

## Author Contributions

**Conceptualization:** Yan-Hua Wu, Pei-Gang Wang, Jing An.

**Data curation:** Yan-Hua Wu, Pei-Gang Wang, Jing An.

**Formal analysis:** Wei Yang, Yan-Hua Wu, Zhi-Hai Qin.

**Funding acquisition:** Zi-Yang Sheng, Pei-Gang Wang, Jing An.

**Investigation:** Wei Yang, Yan-Hua Wu, Shuang-Qing Liu, Zi-Yang Sheng, Zi-Da Zhen, Rui-Qi Gao, Xiao-Yun Cui, Dong-Ying Fan, Zhi-Hai Qin, Ai-Hua Zheng.

**Methodology:** Wei Yang, Yan-Hua Wu, Pei-Gang Wang, Jing An.

**Supervision:** Pei-Gang Wang, Jing An.

**Writing – original draft:** Yan-Hua Wu, Pei-Gang Wang, Jing An.

**Writing – review & editing:** Wei Yang, Yan-Hua Wu, Pei-Gang Wang, Jing An.

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
