## [Decision Letter · Decision Letter 0]

24 Jun 2020

Dear Dr. An,

Thank you very much for submitting your manuscript "S100A4+ Macrophages Assist Zika Virus to Invade and Persist in Seminiferous Tubules via Interferon-gamma Mediation" for consideration at PLOS Pathogens. As with all papers reviewed by the journal, your manuscript was reviewed by members of the editorial board and by several independent reviewers. In light of the reviews (below this email), we would like to invite the resubmission of a significantly-revised version that takes into account the reviewers' comments. 

We cannot make any decision about publication until we have seen the revised manuscript and your response to the reviewers' comments. Your revised manuscript is also likely to be sent to reviewers for further evaluation.

Sincerely,

Mehul Suthar

Associate Editor

PLOS Pathogens

Ted Pierson

Section Editor

PLOS Pathogens

Kasturi Haldar

Editor-in-Chief

PLOS Pathogens

orcid.org/0000-0001-5065-158X

Michael Malim

Editor-in-Chief

PLOS Pathogens

orcid.org/0000-0002-7699-2064

Reviewer's Responses to Questions

**Part I - Summary**

Reviewer #1: The manuscript titled “S100A4+ Macrophages Assist Zika Virus to Invade and Persist in Seminiferous Tubules via Interferon-gamma Mediation” describes the role of S100 A4 macrophages in testicles during ZIKV infection. The mechanism that regulates the ZIKV pathogenesis in infecting sperm cells has not been delineated yet. While the finding shed some new and important knowledge in this particular area, authors should emphasize the significance of their findings in terms of disease prevention.

Reviewer #2: Here, the authors have produced a paper in which the mechanisms of Zika virus testicular invasion and persistence were examined. The authors use a combination of in vivo and in vitro models to show that S1004a+ macrophages significantly contribute to the trafficking of ZIKV to the male reproductive tract and clearly play a crucial role in establishment of ZIKV infection in the testicles. The results are comprehensive, but I believe there are few things that need to be addressed to facilitate understanding by the reader and a few scientific issues that are not currently addressed. First, grammar and syntax need to be improved throughout. Second, there are no line numbers, which make it difficult to call out specific areas that need improvement. Please include line numbers. Specific comments follow.

Reviewer #3: The manuscript by Yang et al. presents data showing that S100A4+ macrophages are a target of ZIKV infection in the testes, where they are recruited. These cells secrete IFNgamma, which downregulates the tight junction protein Claudin-1 in the testes. S100A4+ cells may therefore increase the permeability of the blood-testes barrier. The experiments are performed well and identify a novel cell type infected by ZIKV in the testes.

Reviewer #4: In this study, the authors use A6 mice, AG6 mice and S100a4-/- Ifnar-/- mice to investigate the pathological changes in the ZIKV-infected testis. They report to find a subpopulation of S100A4+ bone marrow-derived macrophages in the testes of ZIKV-infected mice. They also found that S100A4+ macrophages were specifically recruited into interstitial space of the testes and differentiated into interferon-γ expressing M1 macrophages. Although the study question is important, and most of the results are clearly presented, there are several weaknesses in the data interpretation and choice of the animal model.

Major weakness is the choice of animal model. During the early stages of animal model development for Zika, Ifnar-/- mouse were used to study ZIKV pathogenesis including testicular infection. However, very soon it was realized that these immunocompromised mouse models do not mimic human testicular infection of Zika virus. Persistence of ZIKV in human testes is not accompanied with pain or orchitis-like symptoms and severe testicular damage as seen in these AG6 or A6 mouse models. Therefore, any data on severe disruption of BTB and gross pathology seen in these mouse models is not applicable to human ZIKV infection and persistence in the testes and thus lowers the enthusiasm of this study. Recently immunocompetent mouse models including hSTAT2 knock in mice have been established to study ZIKV pathogenesis in placenta and fetal brains. They would be ideal to address the overall objectives of this study. These mice are available at commercial vendors and it will be nice if at least few experiments are repeated with immunocompetent mice to strengthen the relevance of this study.

Other comments:

Fig 1: What is the ZIKV titers in other organs at these time points. Is the virus cleared from other organs and blood by day 14 and ZIKV has established persistence only in the immune privilege compartment of the testes? In the A6 mouse models, it is important to show this data to validate persistence. If virus is present in the blood and other organs at later time points, that will mean that immunocompromised mice cannot clear virus from all organs including testes and will affect data interpretation. The author’s previous AG6 data demonstrates severe damage of the testes while symptoms of testicular damage like orchitis and pain are not reported in humans. Since Fig 1 experiments use A6 mice (not AG6), the virus replication in blood and other organs at different time points and survival data should be included here to correlate with testes virus titers and gene expression data. Actually, a comparative figure of virus, and S100A4 and chemokine gene expression in testes and other organs like blood, brain and lung would strengthen the study and determine if the effect is specific to testes or a limitation of this animal model.

Fig 2E: doesn’t make sense- why would the virus titers go down at 36 hrs and then go up again? Was the virus at 24 hrs residual virus after infection? This needs clarification

Fig 4: By day 14 the boundaries between seminiferous tubules (ST) and interstitium appears to be totally lost and the testicular damage is extensive. Therefore, it cannot be said that the virus infected cells diffused into ST. At that time, there is no distinct interstitium and ST lumen.

Last sentence of Fig 4 results-‘At late stage when spermatogenic cells diminished, luminal S100A4+ macrophages replaced them to support the infection of ZIKV. In this way, ZIKV changed target cells and persisted in testicles for longer time’- Here do the authors mean that ZIKV infected Spermatogonium stem cells die due to infection at later stages and therefore the virus is seen only in macrophages? In that case they should conduct caspase-3 or some cell death assays to determine death of infected spermatogonium cells.

Fig 5A: Again, ZIKV viral load in other organs should be assessed in SA6 mice. Based on Fig 5A, it appears that virus in the blood is high in the A6 mice at day 7 and 14 and correlates with high virus in the testes. What is the virus titer in the blood and testes in both A6 and SA6 mice at day 7, 21 and 28. Does the kinetics of virus clearance in the blood and other organs follow similar pattern as in testes? Is less virus in SA6 mice in the testes at days 10 and 14 due to less virus in the blood? Can a simple interpretation of Fig 5A-J can be that since the virus reaching testes is less in SA6 mice (due to reduced viremia) that leads to reduced replication in the testes and reduced damage to the ST cells and spermatogonium cells.

Although the authors measured IFN-g, since S100A4 macs are pro-inflammatory, it is pertinent to measure the cytokine/chemokine response in the testes using multiplex assays or ELISA (not RT-PCR). The kinetics of S100A4 positive macs and virus titers in the testes and its correlation with cytokines/chemokines produced at different time points and disruption of ST boundaries will be insightful.

**Part II – Major Issues: Key Experiments Required for Acceptance**

Reviewer #1: 1. If the bone marrow derived macrophages are not required during early stage of infection, how does the virus enters and cause infection and how it overcomes the host IFN response to cause an infection?

2. It was previously documented that testicular macrophages can be infected by ZIKV in vitro. These macrophages were reported to have a dual role by triggering host immune response and also increasing the permeability of BTB. According to the current findings, authors should discuss how the myeloid macrophages could overcome the immune response/immune suppressive properties of resident macrophages during ZIKV infection.

3. As this is the first report on role of s100A4 in ZIKV, the authors should discuss the significance of S100 A4 macrophages in any other viral infection (per say) that are sexually transmitted.

4. What would be the rationale for the recruitment of bone marrow macrophages during ZIKV infection; Would it be an unique process to ZIKA?

5. This study lack insights to the possible mechanism by which S100 A4+ cells supports ZIKV infection into seminiferous tubules? It will be good to have this demonstrated.

6. What would have been the role of newly recruited macrophages in sperm cells during cellular stress/tissue injury without ZIKV infection

7. It is essential to show the experiment of infecting S100A4 macrophage with ZIKV ex vivo and introduce the infected cells back to SA6 mice. Does seminiferous tubules invasion occur?

Reviewer #3: 1. The authors say that the S100A4+ cells help ZIKV to evade CD8+ T cells. While they show that the S100A4+ cells are within the seminiferous tubules, while CD8+ T cells are excluded from the seminiferous tubules, this does not show that the S100A4+ cells help ZIKV to evade CD8+ T cells. In the S100A4 knockout mice, were CD8+ T cells increased in the seminiferous tubules?

2. What is the permeability of the testes in S100A4 knockout mice after ZIKV infection?

**Part III – Minor Issues: Editorial and Data Presentation Modifications**

Reviewer #1: 1. What makes the virus attach to the sperm for several months after ZIKV infection when S100 A4 starts decline 21dpi (Fig 2E)

2. Why there is an upregulation of macrophages in spleen during ZIKV. What is their role in tissue tropism?

3. What would be the clinical relevance of serum claudin 1 and ZIKA-can it be used for prognosis?

4. If possible, EM pictures (Fig 4D-I) will be easier to interpret with the same magnifying power and a simplified cartoon showing anatomical location of macrophage invading seminiferous tubules.

5. The comparison between A6 and AG6 susceptibility (IFN-gamma effect) to ZIKV (Fig 7H- J) would be clearer with the arrows to point out the affected cells, the brightness of the staining colors are poorly presented.

Reviewer #2: Introduction:

Aren’t A6 and AG6 mice also IFN-beta deficient?

Results:

It would be helpful to know what timepoint RNAseq was done to understand when S10004a transcript abundance was increased.

Some strains of ZIKV cause rapidly fatal infection in type I interferon deficient mice at 10e4 PFU. Mice here survived until 28 days. Is this unusual? Also, why was d7 chosen as the earliest timepoint? Surely, invasion of the male reproductive tract occurs much earlier after establishment of infection in the periphery.

Figure 1: How was Zika virus infection confirmed in samples that were used for RNAseq? What are the ZIKV viral titers in testis samples at 7, 14, and 28 dpi?

Figure 5: numerous studies have shown clearance of ZIKV from the peripheral blood of type I interferon deficient mice by day 7. Here, you show detectable vRNA loads at 7 and 14 dpi. Is this consistent with previous work with this particular ZIKV strain in mice?

Figure 5: Given that serum and testis viral loads were broadly similar, albeit lower in the SA6 group it would be interesting to know if SA6 have an equivalent capacity to sexually transmit ZIKV and/or how much infectious virus is shed in the semen compared to A6 mice.

In general, why was the IP route chosen for mouse inoculations and could this ultimately impact trafficking to the male reproductive tract compared to a subcutaneous inoculation, for example?

Methods:

More information is needed on the provenance of the virus strain used. What is the exact passage history of the virus isolate and stock used for experiments? Was the stock sequence verified and if so, what differences (if any) exist between it and the GenBank sequence?

Reviewer #3: 1. Some editing of grammar is required.

2. A graphical depiction of the proposed mechanism may be helpful.

Reviewer #4: Minor comments:

Introduction, first para, line 6- sentence requires reformatting and more specific details such as ‘some of them more than 6 months’ is vague- specific percentage or study details should be included.

Line 8 in the same paragraph- ‘because of threat to male health’- please elaborate on the data to support threat to the male health

Testicle is wrong term at many places including Blood-Testicle-Barrier. Testicle generally refers to testes along with epididymis.

Similarly, there are many sentences that need careful reading and either reformatting or inclusion of specific details.

English language and grammar should be carefully edited.

PLOS authors have the option to publish the peer review history of their article (what does this mean?). If published, this will include your full peer review and any attached files.

Reviewer #1: No

Reviewer #2: No

Reviewer #3: No

Reviewer #4: No
---

## [Decision Letter · Decision Letter 1]

1 Oct 2020

Dear Dr. An,

We are pleased to inform you that your manuscript 'S100A4+ Macrophages Facilitate Zika Virus Invasion and Persistence in the Seminiferous Tubules via Interferon-gamma Mediation' has been provisionally accepted for publication in PLOS Pathogens.

Best regards,

Mehul Suthar

Associate Editor

PLOS Pathogens

Ted Pierson

Section Editor

PLOS Pathogens

Kasturi Haldar

Editor-in-Chief

PLOS Pathogens

orcid.org/0000-0001-5065-158X

Michael Malim

Editor-in-Chief

PLOS Pathogens

orcid.org/0000-0002-7699-2064

Reviewer Comments (if any, and for reference):

Reviewer's Responses to Questions

**Part I - Summary**

Reviewer #2: The authors made a majority of the recommended changes requested during initial peer-review of this manuscript and if the changes were not made, a sufficient explanation was provided. The changes significantly enhanced the credibility and scientific nature of the manuscript. The readers of the article can now fully understand the scientific methods used throughout this study and accurately interpret the scientific findings without bias or incomplete information. I recommend that this article should be accepted for publication without additional major changes to the manuscript.

Reviewer #3: The authors have adequately addressed my questions.

Reviewer #4: Th authors have tried to address most of the concerns raised by this reviewer.

However, the new supplemental data (Sup Fig 1 A) validates the concern raised by previous reviewer- The virus in other organs like liver and lung is cleared by day 21. However, virus still remains very high in the blood suggesting that immunodeficient nature of these mice allows uncontrolled virus replication in the periphery that may enter immune privilege sites like brain and testes as well as other tissues. While other organs like liver and lung are able to clear the virus most likely due to body's robust antiviral response, virus lingers in brain and testes due to restricted access of adaptive immune cells in these organs. Therefore, it still appears that the case in brain and testes is not so strong about persistence but rather its about delayed clearance of virus in immune privilege organs.

This issue has to be very clearly explained in the manuscript.

**Part III – Minor Issues: Editorial and Data Presentation Modifications**

Reviewer #2: I do strongly recommend that the authors include the additional details provided in the rebuttal letter about ZIKV strain passage history in the methods section of the text.

Reviewer #4: Th authors could not validate some of their data in more relevant animal model like hSTAT2 mouse. If possible they should include in the discussion that validation of their data in more relevant animal model as part of future studies.

PLOS authors have the option to publish the peer review history of their article (what does this mean?). If published, this will include your full peer review and any attached files.

Reviewer #1: No

Reviewer #2: No

Reviewer #3: No

Reviewer #4: No

---

## [Editor Report · Acceptance letter]

2 Dec 2020

Dear Dr. An,

We are delighted to inform you that your manuscript, "S100A4+ Macrophages Facilitate Zika Virus Invasion and Persistence in the Seminiferous Tubules via Interferon-gamma Mediation," has been formally accepted for publication in PLOS Pathogens.

Best regards,

Kasturi Haldar

Editor-in-Chief

PLOS Pathogens

orcid.org/0000-0001-5065-158X

Michael Malim

Editor-in-Chief

PLOS Pathogens

orcid.org/0000-0002-7699-2064